# On Explore-Then-Commit Strategies

**Aurélien Garivier**[*]
Institut de Mathématiques de Toulouse; UMR5219
Université de Toulouse; CNRS
UPS IMT, F-31062 Toulouse Cedex 9, France
`aurelien.garivier@math.univ-toulouse.fr`

**Emilie Kaufmann**
Univ. Lille, CNRS, Centrale Lille, Inria SequeL
UMR 9189, CRIStAL - Centre de Recherche en Informatique Signal et Automatique de Lille
F-59000 Lille, France
`emilie.kaufmann@univ-lille1.fr`

**Tor Lattimore**
University of Alberta
116 St & 85 Ave, Edmonton, AB T6G 2R3, Canada
`tor.lattimore@gmail.com`

## Abstract

We study the problem of minimising regret in two-armed bandit problems with Gaussian rewards. Our objective is to use this simple setting to illustrate that strategies based on an exploration phase (up to a stopping time) followed by exploitation are necessarily suboptimal. The results hold regardless of whether or not the difference in means between the two arms is known. Besides the main message, we also refine existing deviation inequalities, which allow us to design fully sequential strategies with finite-time regret guarantees that are (a) asymptotically optimal as the horizon grows and (b) order-optimal in the minimax sense. Furthermore we provide empirical evidence that the theory also holds in practice and discuss extensions to non-gaussian and multiple-armed case.

## 1 Introduction

It is now a very frequent issue for companies to optimise their daily profits by choosing between one of two possible website layouts. A natural approach is to start with a period of A/B Testing (exploration) during which the two versions are uniformly presented to users. Once the testing is complete, the company displays the version believed to generate the most profit for the rest of the month (exploitation). The time spent exploring may be chosen adaptively based on past observations, but could also be fixed in advance. Our contribution is to show that strategies of this form are much worse than if the company is allowed to dynamically select which website to display without restrictions for the whole month.

Our analysis focusses on a simple sequential decision problem played over $T$ time-steps. In time-step $t \in 1, 2, \ldots, T$ the agent chooses an action $A_t \in \{1, 2\}$ and receives a normally distributed reward

---

[*]This work was partially supported by the CIMI (Centre International de Mathématiques et d'Informatique) Excellence program while Emilie Kaufmann visited Toulouse in November 2015. The authors acknowledge the support of the French Agence Nationale de la Recherche (ANR), under grants ANR-13-BS01-0005 (project SPADRO) and ANR-13-CORD-0020 (project ALICIA).

$Z_t \sim \mathcal{N}(\mu_{A_t}, 1)$ where $\mu_1, \mu_2 \in \mathbb{R}$ are the unknown mean rewards for actions $1$ and $2$ respectively. The goal is to find a strategy $\pi$ (a way of choosing each action $A_t$ based on past observation) that maximises the cumulative reward over $T$ steps in expectation, or equivalently minimises the regret

$$R_\mu^\pi(T) = T \max\{\mu_1, \mu_2\} - \mathbb{E}_\mu \left[ \sum_{t=1}^{T} \mu_{A_t} \right] . \tag{1}$$

This framework is known as the multi-armed bandit problem, which has many applications and has been studied for almost a century [Thompson, 1933]. Although this setting is now quite well understood, the purpose of this article is to show that strategies based on distinct phases of exploration and exploitation are necessarily suboptimal. This is an important message because exploration followed by exploitation is the most natural approach and is often implemented in applications (including the website optimisation problem described above). Moreover, strategies of this kind have been proposed in the literature for more complicated settings [Auer and Ortner, 2010, Perchet and Rigollet, 2013, Perchet et al., 2015]. Recent progress on optimal exploration policies (e.g., by Garivier and Kaufmann [2016]) could have suggested that well-tuned variants of two-phase strategies might be near-optimal. We show, on the contrary, that optimal strategies for multi-armed bandit problems *must* be fully-sequential, and in particular should mix exploration and exploitation. It is known since the work of Wald [1945] on simple hypothese testing that sequential procedures can lead to significant gains. Here, the superiority of fully sequential procedures is consistent with intuition: if one arm first appears to be better, but if subsequent observations are disappointing, the obligation to commit at some point can be restrictive. In this paper, we give a crisp and precise description of how restrictive it is: it leads to regret asymptotically twice as large on average. The proof of this result combines some classical techniques of sequential analysis and of the bandit literature.

We study two settings, one when the gap $\Delta = |\mu_1 - \mu_2|$ is known and the other when it is not. The most straight-forward strategy in the former case is to explore each action a fixed number of times $n$ and subsequently exploit by choosing the action that appeared best while exploring. It is easy to calculate the optimal $n$ and consequently show that this strategy suffers a regret of $R_\mu^\pi(T) \sim 4 \log(T)/\Delta$. A more general approach is to use a so-called *Explore-Then-Commit* (ETC) strategy, following a nomenclature introduced by Perchet et al. [2015]. An ETC strategy explores each action alternately until some data-dependent stopping time and subsequently commits to a single action for the remaining time-steps. We show in Theorem 2 that by using a sequential probability ratio test (SPRT) it is possible to design an ETC strategy for which $R_\mu^\pi(T) \sim \log(T)/\Delta$, which improves on the above result by a factor of $4$. We also prove a lower bound showing that no ETC strategy can improve on this result. Surprisingly it is possible to do even better by using a fully sequential strategy inspired by the UCB algorithm for multi-armed bandits [Katehakis and Robbins, 1995]. We design a new strategy for which $R_\mu^\pi(T) \sim \log(T)/(2\Delta)$, which improves on the fixed-design strategy by a factor of $8$ and on SPRT by a factor of $2$. Again we prove a lower bound showing that no strategy can improve on this result.

For the case where $\Delta$ is unknown, fixed-design strategies are hopeless because there is no reasonable tuning for the exploration budget $n$. However, it is possible to design an ETC strategy for unknown gaps. Our approach uses a modified fixed-budget best arm identification (BAI) algorithm in its exploration phase (see e.g., Even-Dar et al. [2006], Garivier and Kaufmann [2016]) and chooses the recommended arm for the remaining time-steps. In Theorem 5 we show that a strategy based on this idea satisfies $R_\mu^\pi(T) \sim 4 \log(T)/\Delta$, which again we show is optimal within the class of ETC strategies. As before, strategies based on ETC are suboptimal by a factor of $2$ relative to the optimal rates achieved by fully sequential strategies such as UCB, which satisfies $R_\mu^\pi(T) \sim 2 \log(T)/\Delta$ [Katehakis and Robbins, 1995].

In a nutshell, strategies based on fixed-design or ETC are necessarily suboptimal. That this failure occurs even in the simple setting considered here is a strong indicator that they are suboptimal in more complicated settings. Our main contribution, presented in more details in Section 2, is to fully characterise the achievable asymptotic regret when $\Delta$ is either known or unknown and the strategies are either fixed-design, ETC or fully sequential. All upper bounds have explicit finite-time forms, which allow us to derive optimal minimax guarantees. For the lower bounds we give a novel and generic proof of all results. All proofs contain new, original ideas that we believe are fundamental to the understanding of sequential analysis.

## 2 Notation and Summary of Results

We assume that the horizon $T$ is known to the agent. The optimal action is $a^* = \arg\max(\mu_1, \mu_2)$, its mean reward is $\mu^* = \mu_{a^*}$, and the gap between the means is $\Delta = |\mu_1 - \mu_2|$. Let $\mathcal{H} = \mathbb{R}^2$ be the set of all possible pairs of means, and $\mathcal{H}_\Delta = \{\mu \in \mathbb{R}^2 : |\mu_1 - \mu_2| = \Delta\}$. For $i \in \{1, 2\}$ and $n \in \mathbb{N}$ let $\hat{\mu}_{i,n}$ be the empirical mean of the $i$th action based on the first $n$ samples. Let $A_t$ be the action chosen in time-step $t$ and $N_i(t) = \sum_{s=1}^{t} \mathbb{1}\{A_s = i\}$ be the number of times the $i$th action has been chosen after time-step $t$. We denote by $\hat{\mu}_i(t) = \hat{\mu}_{i,N_i(t)}$ the empirical mean of the $i$th arm after time-step $t$.

A strategy is denoted by $\pi$, which is a function from past actions/rewards to a distribution over the next actions. An ETC strategy is governed by a sampling rule (which determines which arm to sample at each step), a stopping rule (which specifies when to stop the exploration phase) and a decision rule indicating which arm is chosen in the exploitation phase. As we consider two-armed, Gaussian bandits with equal variances, we focus here on uniform sampling rules, which have been shown in Kaufmann et al. [2014] to be optimal in that setting. For this reason, we define an ETC strategy as a pair $(\tau, \hat{a})$, where $\tau$ is an even stopping time with respect to the filtration $(\mathcal{F}_t = \sigma(Z_1, \ldots, Z_t))_t$ and $\hat{a} \in \{1, 2\}$ is $\mathcal{F}_\tau$-measurable. In all the ETC strategies presented in this paper, the stopping time $\tau$ depends on the horizon $T$ (although this is not reflected in the notation). At time $t$, the action picked by the ETC strategy is $A_t = \begin{cases} 1 & \text{if } t \leq \tau \text{ and } t \text{ is odd}, \\ 2 & \text{if } t \leq \tau \text{ and } t \text{ is even}, \\ \hat{a} & \text{otherwise}. \end{cases}$

The regret for strategy $\pi$, given in Eq. (1), depends on $T$ and $\mu$. Assuming, for example that $\mu_1 = \mu_2 + \Delta$, then an ETC strategy $\pi$ chooses the suboptimal arm $N_2(T) = \frac{\tau \wedge T}{2} + (T - \tau)_+ \mathbb{1}\{\hat{a} = 2\}$ times, and the regret $R_\mu^\pi(T) = \Delta\mathbb{E}_\mu[N_2(T)]$ thus satisfies

$$\Delta\mathbb{E}_\mu[(\tau \wedge T)/2] \leq R_\mu^\pi(T) \leq (\Delta/2)\mathbb{E}_\mu[\tau \wedge T] + \Delta T\, \mathbb{P}_\mu(\tau \leq T, \hat{a} \neq a^*). \quad (2)$$

We denote the set of all ETC strategies by $\Pi_{\text{ETC}}$. A fixed-design strategy is and ETC strategy for which there exists an integer $n$ such that $\tau = 2n$ almost surely, and the set of all such strategies is denoted by $\Pi_{\text{DETC}}$. The set of all strategies is denoted by $\Pi_{\text{ALL}}$. For $\mathcal{S} \in \{\mathcal{H}, \mathcal{H}_\Delta\}$, we are interested in strategies $\pi$ that are uniformly efficient on $\mathcal{S}$, in the sense that

$$\forall \mu \in \mathcal{S}, \forall \alpha > 0,\ R_\mu^\pi(T) = o(T^\alpha). \quad (3)$$

We show in this paper that any uniformly efficient strategy in $\Pi$ has a regret at least equal to $C_{\mathcal{S}}^\Pi \log(T)/|\mu_1 - \mu_2|(1 - o_T(1))$ for every parameter $\mu \in \mathcal{S}$, where $C_{\mathcal{S}}^\Pi$ is given in the adjacent table. Furthermore, we prove that these results are tight. In each case, we propose a uniformly efficient strategy matching

|  | $\Pi_{\text{ALL}}$ | $\Pi_{\text{ETC}}$ | $\Pi_{\text{DETC}}$ |
|---|---|---|---|
| $\mathcal{H}$ | 2 | 4 | NA |
| $\mathcal{H}_\Delta$ | 1/2 | 1 | 4 |

this bound. In addition, we prove a tight and non-asymptotic regret bound which also implies, in particular, minimax rate-optimality.

The paper is organised as follows. First we consider ETC and fixed-design strategies when $\Delta$ known and unknown (Section 3). We then analyse fully sequential strategies that interleave exploration and exploitation in an optimal way (Section 4). For known $\Delta$ we present a novel algorithm that exploits the additional information to improve the regret. For unknown $\Delta$ we briefly recall the well-known results, but also propose a new regret analysis of the UCB* algorithm, a variant of UCB that can be traced back to Lai [1987], for which we also obtain order-optimal minimax regret. Numerical experiments illustrate and empirically support our results in Section 5. We conclude with a short discussion on non-uniform exploration, and on models with more than 2 arms, possibly non Gaussian. All the proofs are given in the supplementary material. In particular, our simple, unified proof for all the lower bounds is given in Appendix A.

## 3 Explore-Then-Commit Strategies

**Fixed Design Strategies for Known Gaps.** As a warm-up we start with the fixed-design ETC setting where $\Delta$ is known and where the agent chooses each action $n$ times before committing for the remainder.

The optimal decision rule is obviously $\hat{a} = \arg\max_i \hat{\mu}_{i,n}$ with ties broken arbitrarily. The formal description of the strategy is given in Algorithm 1, where $W$ denotes the Lambert function implicitly defined for $y > 0$ by $W(y)\exp(W(y)) = y$. We denote the regret associated to the choice of $n$ by $R_\mu^n(T)$. The following theorem is not especially remarkable except that the bound is sufficiently refined to show certain negative lower-order terms that would otherwise not be apparent.

```
input: T and Δ
n := ⌈2W(T²Δ⁴/(32π))/Δ²⌉
for k ∈ {1, ..., n} do
    choose A_{2k-1} = 1 and A_{2k} = 2
end for
â := arg max_i μ̂_{i,n}
for t ∈ {2n + 1, ..., T} do
    choose A_t = â
end for
```

**Algorithm 1:** FB-ETC algorithm

**Theorem 1.** *Let $\mu \in \mathcal{H}_\Delta$, and let*

$$\overline{n} = \left\lceil \frac{2}{\Delta^2} W\left(\frac{T^2\Delta^4}{32\pi}\right)\right\rceil . \quad \text{Then} \quad R_\mu^{\overline{n}}(T) \le \frac{4}{\Delta}\log\left(\frac{T\Delta^2}{4.46}\right) - \frac{2}{\Delta}\log\log\left(\frac{T\Delta^2}{4\sqrt{2\pi}}\right) + \Delta$$

*whenever $T\Delta^2 > 4\sqrt{2\pi e}$, and $R_\mu^{\overline{n}}(T) \le T\Delta/2 + \Delta$ otherwise. In all cases, $R_\mu^{\overline{n}}(T) \le 2.04\sqrt{T} + \Delta$. Furthermore, for all $\varepsilon > 0, T \ge 1$ and $n \le 4(1 - \varepsilon)\log(T)/\Delta^2$,*

$$R_\mu^n(T) \ge \left(1 - \frac{2}{n\Delta^2}\right)\left(1 - \frac{8\log(T)}{\Delta^2 T}\right)\frac{\Delta T^\varepsilon}{2\sqrt{\pi\log(T)}} .$$

*As $R_\mu^n(T) \ge n\Delta$, this entails that $\inf_{1\le n\le T} R_\mu^n(T) \sim 4\log(T)/\Delta$.*

The proof of Theorem 1 is in Appendix B. Note that the "asymptotic lower bound" $4\log(T)/\Delta$ is actually not a lower bound, even up to an additive constant: $R_\mu^{\overline{n}}(T) - 4\log(T)/\Delta \to -\infty$ when $T \to \infty$. Actually, the same phenomenon applies many other cases, and it should be no surprise that, in numerical experiments, some algorithm reach a regret smaller than Lai and Robbins asymptotic lower bound, as was already observed in several articles (see e.g. Garivier et al. [2016]). Also note that the term $\Delta$ at the end of the upper bound is necessary: if $\Delta$ is large, the problem is statistically so simple that one single observation is sufficient to identify the best arm; but that observation cannot be avoided.

**Explore-Then-Commit Strategies for Known Gaps.** We now show the existence of ETC strategies that improve on the optimal fixed-design strategy. Surprisingly, the gain is significant. We describe an algorithm inspired by ideas from hypothesis testing and prove an upper bound on its regret that is minimax optimal and that asymptotically matches our lower bound.

Let $P$ be the law of $X - Y$, where $X$ (resp. $Y$) is a reward from arm 1 (resp. arm 2). As $\Delta$ is known, the exploration phase of an ETC algorithm can be viewed as a statistical test of the hypothesis $H_1 : (P = \mathcal{N}(\Delta, 2))$ against $H_2 : (P = \mathcal{N}(-\Delta, 2))$. The work of Wald [1945] shows that a significant gain in terms of expected number of samples can be obtained by using a sequential rather than a batch test. Indeed, for a batch test, a sample size of $n \sim (4/\Delta^2)\log(1/\delta)$ is necessary to guarantee that both type I and type II errors are upper bounded by $\delta$. In contrast, when a random number of samples is permitted, there exists a sequential probability ratio test (SPRT) with the same guarantees that stops after a random number $N$ of samples with expectation $\mathbb{E}[N] \sim \log(1/\delta)/\Delta^2$ under both $H_1$ and $H_2$. The SPRT stops when the absolute value of the log-likelihood ratio between $H_1$ and $H_2$ exceeds some threshold. Asymptotic upper bound on the expected number of samples used by a SPRT, as well as the (asymptotic) optimality of such procedures among the class of all sequential tests can be found in [Wald, 1945, Siegmund, 1985].

Algorithm 2 is an ETC strategy that explores each action alternately, halting when sufficient confidence is reached according to a SPRT. The threshold depends on the gap $\Delta$ and the horizon $T$ corresponding to a risk of $\delta = 1/(T\Delta^2)$. The exploration phase ends at the stopping time

$$\tau = \inf\left\{t = 2n : \left|\hat{\mu}_{1,n} - \hat{\mu}_{2,n}\right| \ge \frac{\log(T\Delta^2)}{n\Delta}\right\}.$$

If $\tau < T$ then the empirical best arm $\hat{a}$ at time $\tau$ is played until time $T$. If $T\Delta^2 \le 1$, then $\tau = 1$

```
input: T and Δ
A_1 = 1, A_2 = 2, t := 2
while (t/2)Δ|μ̂_1(t) − μ̂_2(t)| < log(TΔ²) do
    choose A_{t+1} = 1 and A_{t+2} = 2,
    t := t + 2
end while
â := arg max_i μ̂_i(t)
while t ≤ T do
    choose A_t = â,
    t := t + 1
end while
```

**Algorithm 2:** SPRT ETC algorithm

(one could even define $\tau = 0$ and pick a random arm). The following theorem gives a non-asymptotic upper bound on the regret of the algorithm. The results rely on non-asymptotic upper bounds on the expectation of $\tau$, which are interesting in their own right.

**Theorem 2.** *If $T\Delta^2 \geq 1$, then the regret of the SPRT-ETC algorithm is upper-bounded as*

$$R_\mu^{\text{SPRT-ETC}}(T) \leq \frac{\log(eT\Delta^2)}{\Delta} + \frac{4\sqrt{\log(T\Delta^2)} + 4}{\Delta} + \Delta \, .$$

*Otherwise it is upper bounded by $T\Delta/2 + \Delta$, and for all $T$ and $\Delta$ the regret is less than $10\sqrt{T/e} + \Delta$.*

The proof of Theorem 2 is given in Appendix C. The following lower bound shows that no uniformly efficient ETC strategy can improve on the asymptotic regret of Algorithm 2. The proof is given in Section A together with the other lower bounds.

**Theorem 3.** *Let $\pi$ be an ETC strategy that is uniformly efficient on $\mathcal{H}_\Delta$. Then for all $\mu \in \mathcal{H}_\Delta$,*

$$\liminf_{T\to\infty} \frac{R_\mu^\pi(T)}{\log(T)} \geq \frac{1}{\Delta} \, .$$

**Explore-Then-Commit Strategies for Unknown Gaps.** When the gap is unknown it is not possible to tune a fixed-design strategy that achieves logarithmic regret. ETC strategies can enjoy logarithmic regret and these are now analysed. We start with the asymptotic lower bound.

**Theorem 4.** *Let $\pi$ be a uniformly efficient ETC strategy on $\mathcal{H}$. For all $\mu \in \mathcal{H}$, if $\Delta = |\mu_1 - \mu_2|$ then*

$$\liminf_{T\to\infty} \frac{R_\mu^\pi(T)}{\log(T)} \geq \frac{4}{\Delta} \, .$$

A simple idea for constructing an algorithm that matches the lower bound is to use a (fixed-confidence) best arm identification algorithm for the exploration phase. Given a risk parameter $\delta$, a $\delta$-PAC BAI algorithm consists of a sampling rule $(A_t)$, a stopping rule $\tau$ and a recommendation rule $\hat{a}$ which is $\mathcal{F}_\tau$ measurable and satisfies, for all $\mu \in \mathcal{H}$ such that $\mu_1 \neq \mu_2$, $\mathbb{P}_\mu(\hat{a} = a^*) \geq 1 - \delta$. In a bandit model with two Gaussian arms, Kaufmann et al. [2014] propose a $\delta$-PAC algorithm using a uniform sampling rule and a stopping rule $\tau_\delta$ that asymptotically attains the minimal sample complexity $\mathbb{E}_\mu[\tau_\delta] \sim (8/\Delta^2) \log(1/\delta)$. Using the regret decomposition (2), it is easy to show that the ETC algorithm using the stopping rule $\tau_\delta$ for $\delta = 1/T$ matches the lower bound of Theorem 4.

Algorithm 3 is a slight variant of this optimal BAI algorithm, based on the stopping time

$$\tau = \inf\left\{ t = 2n : |\hat{\mu}_{1,n} - \hat{\mu}_{2,n}| > \sqrt{\frac{4\log\left(T/(2n)\right)}{n}} \right\}.$$

The motivation for the difference (which comes from a more carefully tuned threshold featuring $\log(T/2n)$ in place of $\log(T)$) is that the confidence level should depend on the unknown gap $\Delta$, which determines the regret when a mis-identification occurs. The improvement only appears in the non-asymptotic regime where we are able to prove both asymptotic optimality and

```
input: T (≥ 3)
A_1 = 1, A_2 = 2, t := 2
while |μ̂_1(t) − μ̂_2(t)| < √(8log(T/t)/t) do
    choose A_{t+1} = 1 and A_{t+2} = 2
    t := t + 2
end while
â := arg max_i μ̂_i(t)
while t ≤ T do
    choose A_t = â
    t := t + 1
end while
```
**Algorithm 3:** BAI-ETC algorithm

order-optimal minimax regret. The latter would not be possible using a fixed-confidence BAI strategy. The proof of this result can be found in Appendix D. The main difficulty is developing a sufficiently strong deviation bound, which we do in Appendix G, and that may be of independent interest. Note that a similar strategy was proposed and analysed by Lai et al. [1983], but in the continuous time framework and with asymptotic analysis only.

**Theorem 5.** *If $T\Delta^2 > 4e^2$, the regret of the BAI-ETC algorithm is upper bounded as*

$$R_\mu^{\text{BAI-ETC}}(T) \leq \frac{4\log\left(\frac{T\Delta^2}{4}\right)}{\Delta} + \frac{334\sqrt{\log\left(\frac{T\Delta^2}{4}\right)}}{\Delta} + \frac{178}{\Delta} + 2\Delta.$$

*It is upper bounded by $T\Delta$ otherwise, and by $32\sqrt{T} + 2\Delta$ in any case.*

# 4 Fully Sequential Strategies for Known and Unknown Gaps

In the previous section we saw that allowing a random stopping time leads to a factor of $4$ improvement in terms of the asymptotic regret relative to the naive fixed-design strategy. We now turn our attention to fully sequential strategies when $\Delta$ is known and unknown. The latter case is the classic 2-armed bandit problem and is now quite well understood. Our modest contribution in that case is the first algorithm that is simultaneously asymptotically optimal and order optimal in the minimax sense. For the former case, we are not aware of any previous research where the gap is known except the line of work by Bubeck et al. [2013], Bubeck and Liu [2013], where different questions are treated. In both cases we see that fully sequential strategies improve on the best ETC strategies by a factor of 2.

**Known Gaps.** We start by stating the lower bound (proved in Section A), which is a straightforward generalisation of Lai and Robbins' lower bound.

**Theorem 6.** *Let $\pi$ be a strategy that is uniformly efficient on $\mathcal{H}_\Delta$. Then for all $\mu \in \mathcal{H}_\Delta$,*

$$\liminf_{T \to \infty} \frac{R_\mu^\pi(T)}{\log T} \geq \frac{1}{2\Delta}$$

We are not aware of any existing algorithm matching this lower bound, which motivates us to introduce a new strategy called $\Delta$-UCB that exploits the knowledge of $\Delta$ to improve the performance of UCB. In each round the algorithm chooses the arm that has been played most often so far unless the other arm has an upper confidence bound that is close to $\Delta$ larger than the empirical estimate of the most played arm. Like ETC strategies, $\Delta$-UCB is not anytime in the sense that it requires the knowledge of both the horizon $T$ and the gap $\Delta$.

---

1: **input:** $T$ and $\Delta$
2: $\varepsilon_T = \Delta \log^{-\frac{1}{8}}(e + T\Delta^2)/4$
3: **for** $t \in \{1, \ldots, T\}$ **do**
4:    let $A_{t,\min} := \arg\min_{i \in 1,2} N_i(t-1)$ and $A_{t,\max} = 3 - A_{t,\min}$
5:    **if** $\hat{\mu}_{A_{t,\min}}(t-1) + \sqrt{\dfrac{2 \log\left(\frac{T}{N_{A_{t,\min}}(t-1)}\right)}{N_{A_{t,\min}}(t-1)}} \geq \hat{\mu}_{A_{t,\max}}(t-1) + \Delta - 2\varepsilon_T$ **then**
6:        choose $A_t = A_{t,\min}$
7:    **else**
8:        choose $A_t = A_{t,\max}$
9:    **end if**
10: **end for**

---

**Algorithm 4:** $\Delta$-UCB

**Theorem 7.** *If $T(2\Delta - 3\varepsilon_T)^2 \geq 2$ and $T\varepsilon_T^2 \geq e^2$, the regret of the $\Delta$-UCB algorithm is upper bounded as*

$$R_\mu^{\Delta\text{-}UCB}(T) \leq \frac{\log\left(2T\Delta^2\right)}{2\Delta(1 - 3\varepsilon_T/(2\Delta))^2} + \frac{\sqrt{\pi \log\left(2T\Delta^2\right)}}{2\Delta(1 - 3\varepsilon_T/\Delta)^2}$$

$$+ \Delta \left[\frac{30e\sqrt{\log(\varepsilon_T^2 T)}}{\varepsilon_T^2} + \frac{80}{\varepsilon_T^2} + \frac{2}{(2\Delta - 3\varepsilon_T)^2}\right] + 5\Delta.$$

*Moreover* $\limsup_{T \to \infty} R_\mu^{\Delta\text{-}UCB}(T)/\log(T) \leq (2\Delta)^{-1}$ *and* $\forall \mu \in \mathcal{H}_\Delta$, $R_\mu^{\Delta\text{-}UCB}(T) \leq 328\sqrt{T} + 5\Delta$.

The proof may be found in Appendix E.

**Unknown Gaps.** In the classical bandit setting where $\Delta$ is unknown, UCB by Katehakis and Robbins [1995] is known to be asymptotically optimal: $R_\mu^{\text{UCB}}(T) \sim 2\log(T)/\Delta$, which matches the lower bound of Lai and Robbins [1985]. Non-asymptotic regret bounds are given for example by Auer et al. [2002], Cappé et al. [2013]. Unfortunately, UCB is not optimal in the minimax sense, which is so far only achieved by algorithms that are not asymptotically optimal [Audibert and Bubeck, 2009, Lattimore, 2015]. Here, with only two arms, we are able to show that Algorithm 5 below is

simultaneously minimax order-optimal and asymptotically optimal. The strategy is essentially the same as suggested by Lai [1987], but with a fractionally smaller confidence bound. The proof of Theorem 8 is given in Appendix F. Empirically the smaller confidence bonus used by UCB* leads to a significant improvement relative to UCB.

---

1: **input:** $T$
2: **for** $t \in \{1, \ldots, T\}$ **do**

3: $\qquad A_t = \underset{i \in \{1,2\}}{\arg \max} \, \hat{\mu}_i(t-1) + \sqrt{\dfrac{2}{N_i(t-1)} \log \left( \dfrac{T}{N_i(t-1)} \right)}$

4: **end for**

---

**Algorithm 5:** UCB*

**Theorem 8.** *For all $\varepsilon \in (0, \Delta)$, if $T(\Delta - \varepsilon)^2 \geq 2$ and $T\varepsilon^2 \geq e^2$, the regret of the UCB* strategy is upper bounded as*

$$R_\mu^{UCB^*}(T) \leq \frac{2 \log \left( \frac{T\Delta^2}{2} \right)}{\Delta \left( 1 - \frac{\varepsilon}{\Delta} \right)^2} + \frac{2 \sqrt{\pi \log \left( \frac{T\Delta^2}{2} \right)}}{\Delta \left( 1 - \frac{\varepsilon}{\Delta} \right)^2} + \Delta \left( \frac{30 e \sqrt{\log(\varepsilon^2 T)} + 16e}{\varepsilon^2} \right) + \frac{2}{\Delta \left( 1 - \frac{\varepsilon}{\Delta} \right)^2} + \Delta.$$

*Moreover,* $\limsup_{T \to \infty} R_\mu^\pi(T) / \log(T) = 2/\Delta$ *and for all* $\mu \in \mathcal{H}$, $R_\mu^\pi(T) \leq 33\sqrt{T} + \Delta$.

Note that if there are $K > 2$ arms, then the strategy above is still asymptotically optimal, but suffers a minimax regret of $\Omega(\sqrt{TK \log(K)})$, which is a factor of $\sqrt{\log(K)}$ suboptimal.

## 5 Numerical Experiments

We represent here the regret of the five strategies presented in this article on a bandit problem with $\Delta = 1/5$, for different values of the horizon. The regret is estimated by $4.10^5$ Monte-Carlo replications. In the legend, the estimated slopes of $\Delta R^\pi(T)$ (in logarithmic scale) are indicated after the policy names.

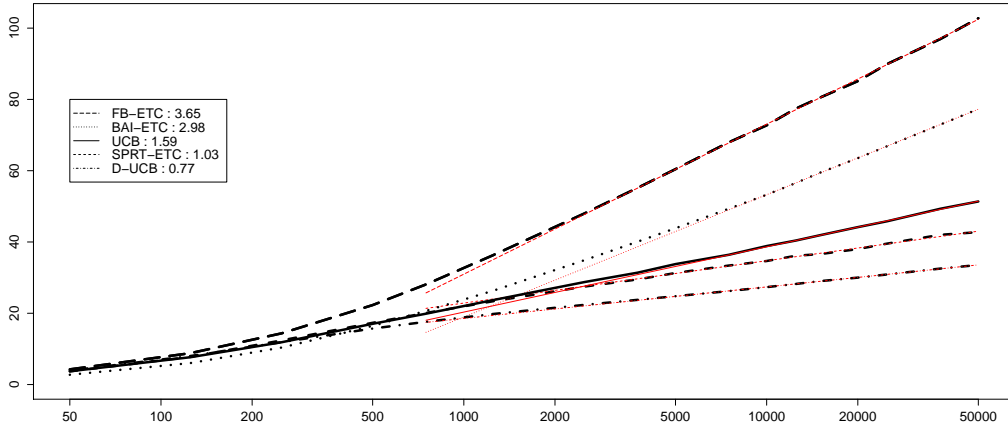

The experimental behavior of the algorithms reflects the theoretical results presented above: the regret asymptotically grows as the logarithm of the horizon, the experimental coefficients correspond approximately to theory, and the relative ordering of the policies is respected. However, it should be noted that for short horizons the hierarchy is not quite the same, and the growth rate is not logarithmic; this question is raised in Garivier et al. [2016]. In particular, on short horizons the Best-Arm Identification procedure performs very well with respect to the others, and starts to be beaten (even by the gap-aware strategies) only when $T\Delta^2$ is much larger that 10.

# 6 Conclusion: Beyond Uniform Exploration, Two Arms and Gaussian distributions

It is worth emphasising the impossibility of non-trivial lower bounds on the regret of ETC strategies using any possible (non-uniform) sampling rule. Indeed, using UCB as a sampling rule together with an a.s. infinite stopping rule defines an artificial but formally valid ETC strategy that achieves the best possible rate for general strategies. This strategy is not a faithful counter-example to our claim that ETC strategies are sub-optimal, because UCB is not a satisfying exploration rule. If exploration is the objective, then uniform sampling is known to be optimal in the two-armed Gaussian case [Kaufmann et al., 2014], which justifies the uniform sampling assumption.

The use of ETC strategies for regret minimisation (e.g., as presented by Perchet and Rigollet [2013]) is certainly not limited to bandit models with 2 arms. The extension to multiple arms is based on the successive elimination idea in which a set of active arms is maintained with arms chosen according to a round robin within the active set. Arms are eliminated from the active set once their optimality becomes implausible and the exploration phase terminates when the active set contains only a single arm (an example is by Auer and Ortner [2010]). The Successive Elimination algorithm has been introduced by Even-Dar et al. [2006] for best-arm identification in the fixed-confidence setting. It was shown to be rate-optimal, and thus a good compromise for both minimizing regret and finding the best arm. If one looks more precisely at mutliplicative constants, however, Garivier and Kaufmann [2016] showed that it is suboptimal for the best arm identification task in almost all settings except two-armed Gaussian bandits. Regarding regret minimization, the present paper shows that it is sub-optimal by a factor 2 on every two-armed Gaussian problem.

It is therefore interesting to investigate the performance in terms of regret of an ETC algorithm using an optimal BAI algorithm. This is actually possible not only for Gaussian distributions, but more generally for one-parameter exponential families, for which Garivier and Kaufmann [2016] propose the asymptotically optimal Track-and-Stop strategy. Denoting $d(\mu, \mu') = \mathrm{KL}(\nu_\mu, \nu_{\mu'})$ the Kullback-Leibler divergence between two distributions parameterised by $\mu$ and $\mu'$, they provide results which can be adapted to obtain the following bound.

**Proposition 1.** *For $\mu$ such that $\mu_1 > \max_{a \neq 1} \mu_a$, the regret of the ETC strategy using Track-and-Stop exploration with risk $1/T$ satisfies*

$$\limsup_{T \to \infty} \frac{R_\mu^{\mathrm{TaS}}(T)}{\log T} \leq T^*(\mu) \left( \sum_{a=2}^{K} w_a^*(\mu)(\mu_1 - \mu_a) \right),$$

*where $T^*(\mu)$ (resp. $w^*(\mu)$) is the the maximum (resp. maximiser) of the optimisation problem*

$$\max_{w \in \Sigma_K} \inf_{a \neq 1} \left[ w_1 d \left( \mu_1, \frac{w_1 \mu_1 + w_a \mu_a}{w_1 + w_a} \right) + w_a d \left( \mu_a, \frac{w_a \mu_1 + w_a \mu_a}{w_1 + w_a} \right) \right],$$

*where $\Sigma_K$ is the set of probability distributions on $\{1, \ldots, K\}$.*

In general, it is not easy to quantify the difference to the lower bound of Lai and Robbins

$$\liminf_{T \to \infty} \frac{R_\mu^\pi(T)}{\log T} \geq \sum_{a=2}^{K} \frac{\mu_1 - \mu_a}{d(\mu_a, \mu_1)}.$$

Even for Gaussian distributions, there is no general closed-form formula for $T^*(\mu)$ and $w^*(\mu)$ except when $K = 2$. However, we conjecture that the worst case is when $\mu_1$ and $\mu_2$ are much larger than the other means: then, the regret is almost the same as in the 2-arm case, and ETC strategies are suboptimal by a factor 2. On the other hand, the most favourable case (in terms of relative efficiency) seems to be when $\mu_2 = \cdots = \mu_K$: then

$$w_1^*(\mu) = \frac{\sqrt{K-1}}{K-1+\sqrt{K-1}}, \qquad w_2^*(\mu) = \cdots = w_K^*(\mu) = \frac{1}{K-1+\sqrt{K-1}}$$

and $T^* = 2(\sqrt{K-1}+1)^2/\Delta^2$, leading to

$$\limsup_{T \to \infty} \frac{R_\mu^{\mathrm{TaS}}(T)}{\log(T)} \leq \left( 1 + \frac{1}{\sqrt{K-1}} \right) \frac{2(K-1)}{\Delta},$$

while Lai and Robbins' lower bound yields $2(K-1)/\Delta$. Thus, the difference grows with $K$ as $2\sqrt{K-1}\log(T)/\Delta$, but the relative difference decreases.

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
