[Supplementary Material · GKL16supp.pdf]

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

> $n := \left\lceil 2W\left(T^2\Delta^4/(32\pi)\right)/\Delta^2 \right\rceil$
> **for** $k \in \{1, \ldots, n\}$ **do**
>      choose $A_{2k-1} = 1$ and $A_{2k} = 2$
> **end for**
> $\hat{a} := \arg\max_i \hat{\mu}_{i,n}$
> **for** $t \in \{2n + 1, \ldots, T\}$ **do**
>      choose $A_t = \hat{a}$
> **end for**

**Algorithm 1:** FB-ETC algorithm

**Theorem 1.** *Let $\mu \in \mathcal{H}_\Delta$, and let*

$$\overline{n} = \left\lceil \frac{2}{\Delta^2} W\left(\frac{T^2\Delta^4}{32\pi}\right) \right\rceil . \quad \textit{Then} \quad R_\mu^{\overline{n}}(T) \leq \frac{4}{\Delta}\log\left(\frac{T\Delta^2}{4.46}\right) - \frac{2}{\Delta}\log\log\left(\frac{T\Delta^2}{4\sqrt{2\pi}}\right) + \Delta$$

*whenever $T\Delta^2 > 4\sqrt{2\pi e}$, and $R_\mu^{\overline{n}}(T) \leq T\Delta/2 + \Delta$ otherwise. In all cases, $R_\mu^{\overline{n}}(T) \leq 2.04\sqrt{T} + \Delta$. Furthermore, for all $\varepsilon > 0, T \geq 1$ and $n \leq 4(1-\varepsilon)\log(T)/\Delta^2$,*

$$R_\mu^n(T) \geq \left(1 - \frac{2}{n\Delta^2}\right)\left(1 - \frac{8\log(T)}{\Delta^2 T}\right)\frac{\Delta T^\varepsilon}{2\sqrt{\pi\log(T)}} .$$

*As $R_\mu^n(T) \geq n\Delta$, this entails that $\inf_{1 \leq n \leq T} R_\mu^n(T) \sim 4\log(T)/\Delta$.*

The proof of Theorem 1 is in Appendix B. Note that the "asymptotic lower bound" $4\log(T)/\Delta$ is actually not a lower bound, even up to an additive constant: $R_\mu^{\overline{n}}(T) - 4\log(T)/\Delta \to -\infty$ when $T \to \infty$. Actually, the same phenomenon applies many other cases, and it should be no surprise that, in numerical experiments, some algorithm reach a regret smaller than Lai and Robbins asymptotic lower bound, as was already observed in several articles (see e.g. Garivier et al. [2016]). Also note that the term $\Delta$ at the end of the upper bound is necessary: if $\Delta$ is large, the problem is statistically so simple that one single observation is sufficient to identify the best arm; but that observation cannot be avoided.

**Explore-Then-Commit Strategies for Known Gaps.** We now show the existence of ETC strategies that improve on the optimal fixed-design strategy. Surprisingly, the gain is significant. We describe an algorithm inspired by ideas from hypothesis testing and prove an upper bound on its regret that is minimax optimal and that asymptotically matches our lower bound.

Let $P$ be the law of $X - Y$, where $X$ (resp. $Y$) is a reward from arm 1 (resp. arm 2). As $\Delta$ is known, the exploration phase of an ETC algorithm can be viewed as a statistical test of the hypothesis $H_1 : (P = \mathcal{N}(\Delta, 2))$ against $H_2 : (P = \mathcal{N}(-\Delta, 2))$. The work of Wald [1945] shows that a significant gain in terms of expected number of samples can be obtained by using a sequential rather than a batch test. Indeed, for a batch test, a sample size of $n \sim (4/\Delta^2)\log(1/\delta)$ is necessary to guarantee that both type I and type II errors are upper bounded by $\delta$. In contrast, when a random number of samples is permitted, there exists a sequential probability ratio test (SPRT) with the same guarantees that stops after a random number $N$ of samples with expectation $\mathbb{E}[N] \sim \log(1/\delta)/\Delta^2$ under both $H_1$ and $H_2$. The SPRT stops when the absolute value of the log-likelihood ratio between $H_1$ and $H_2$ exceeds some threshold. Asymptotic upper bound on the expected number of samples used by a SPRT, as well as the (asymptotic) optimality of such procedures among the class of all sequential tests can be found in [Wald, 1945, Siegmund, 1985].

Algorithm 2 is an ETC strategy that explores each action alternately, halting when sufficient confidence is reached according to a SPRT. The threshold depends on the gap $\Delta$ and the horizon $T$ corresponding to a risk of $\delta = 1/(T\Delta^2)$. The exploration phase ends at the stopping time

$$\tau = \inf\left\{t = 2n : \left|\hat{\mu}_{1,n} - \hat{\mu}_{2,n}\right| \geq \frac{\log(T\Delta^2)}{n\Delta}\right\}.$$

If $\tau < T$ then the empirical best arm $\hat{a}$ at time $\tau$ is played until time $T$. If $T\Delta^2 \leq 1$, then $\tau = 1$

> **input:** $T$ and $\Delta$
> $A_1 = 1, A_2 = 2, t := 2$
> **while** $(t/2)\Delta \left|\hat{\mu}_1(t) - \hat{\mu}_2(t)\right| < \log\left(T\Delta^2\right)$ **

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

## Notation for the Proofs

We denote by $(X_s)$ and $(Y_s)$ the sequence of successive observations from arm 1 and arm 2, so that

$$\hat{\mu}_{1,s} = \frac{1}{s} \sum_{i=1}^{s} X_i \quad \text{and} \quad \hat{\mu}_{2,s} = \frac{1}{s} \sum_{i=1}^{s} Y_i$$

In the proofs of all regret upper bounds given in Appendix B to F we assume without loss of generality that $\mu$ is such that $\mu_1 > \mu_2$ and let $\Delta = \mu_1 - \mu_2$.

## A Proof of the Lower Bounds (Theorems 3, 4, 6 and Lai&Robbins)

Let $\pi$ be a uniformly efficient strategy on some class $\mathcal{S}$, as defined in (3), and let $\lambda \in \mathcal{H}$. If $m(\lambda) = \arg\min\{\lambda_1, \lambda_2\}$, as $\mathbb{E}_\lambda[R_\mu^\pi(T)] = |\lambda_1 - \lambda_2|\mathbb{E}_\lambda[N_{m(\lambda)}(T)]$ this implies in particular that

$$\forall \alpha \in ]0,1], \ \mathbb{E}_\lambda[N_{m(\lambda)}(T)] = o(T^\alpha).$$

Without loss of generality we assume that $\mu_1 = \mu_2 + \Delta$ with $\Delta > 0$. All the lower bounds are based on a change of measure argument, which involves considering an alternative reward vector $(\mu_1', \mu_2')$ that is "not too far" from $(\mu_1, \mu_2)$, but for which the expected behaviour of the algorithm is very different. This is the same approach used by Lai and Robbins [1985], but rewritten and generalised in a more powerful way (in particular regarding the ETC strategies). The improvements come thanks to Inequality 4 in [Garivier et al., 2016], which states that for every $(\mu_1', \mu_2') \in \mathcal{H}$ and for every stopping time $\sigma$ such that $N_2(T)$ is $\mathcal{F}_\sigma$-measurable,

$$\mathbb{E}_\mu\big[N_1(\sigma)\big] \frac{(\mu_1' - \mu_1)^2}{2} + \mathbb{E}_\mu\big[N_2(\sigma)\big] \frac{(\mu_2' - \mu_2)^2}{2} \geq \mathrm{kl}\left(\mathbb{E}_\mu\left[\frac{N_2(T)}{T}\right], \mathbb{E}_{\mu'}\left[\frac{N_2(T)}{T}\right]\right),$$

where $\mathrm{kl}(p,q)$ is the relative entropy between Bernoulli distributions with parameters $p, q \in [0,1]$ respectively. Since $\mathrm{kl}(p,q) \geq (1-p)\log(1/(1-q)) - \log(2)$ for all $p, q \in (0,1)$, one obtains

$$\mathbb{E}_\mu\big[N_1(\sigma)\big] \frac{(\mu_1' - \mu_1)^2}{2} + \mathbb{E}_\mu\big[N_2(\sigma)\big] \frac{(\mu_2' - \mu_2)^2}{2}$$
$$\geq \left(1 - \frac{\mathbb{E}_\mu[N_2(T)]}{T}\right) \log\left(\frac{T}{\mathbb{E}_{\mu'}[N_1(T)]}\right) - \log(2). \tag{4}$$

For $\mu' \in \mathcal{S}$ such that $\mu_1' < \mu_2'$, $\mathbb{E}_\mu[N_2(T)] = o(T^\alpha)$ and $\mathbb{E}_{\mu'}[N_1(T)] = o(T^\alpha)$ for all $\alpha \in ]0,1]$, thus

$$\liminf_{T \to \infty} \frac{\mathbb{E}_\mu\big[N_1(\sigma)\big](\mu_1' - \mu_1)^2/2 + \mathbb{E}_\mu\big[N_2(\sigma)\big](\mu_2' - \mu_2)^2/2}{\log T} \geq 1 .$$

Let us now draw the conclusions in each setting. Observe that while this argument is now routine for general policies, we show here how to apply it very nicely to ETC strategies as well.

**Known gap, General strategy:** $\mathcal{S} = \mathcal{H}_\Delta$. By choosing $\sigma = T$, $\mu_1' = \mu_1$ and $\mu_2' = \mu_1 + \Delta = \mu_2 + 2\Delta$, we obtain

$$\liminf_{T \to \infty} \frac{\mathbb{E}_\mu\big[N_2(T)\big]}{\log T} \geq \frac{1}{(2\Delta)^2/2} .$$

**Unknown gap, General strategy:** $\mathcal{S} = \mathcal{H}$. We use the same choices, except $\mu_2' = \mu_1 + \varepsilon$ for some $\varepsilon > 0$:

$$\liminf_{T \to \infty} \frac{\mathbb{E}_\mu\big[N_2(T)\big]}{\log T} \geq \frac{1}{(\Delta + \varepsilon)^2/2} .$$

**Known gap, ETC strategy:** $\mathcal{S} = \mathcal{H}_\Delta$. For an ETC strategy $\pi$ with a stopping rule $\tau$, one has $N_1(\tau \wedge T) = N_2(\tau \wedge T) = (\tau \wedge T)/2$. Besides, $N_2(T)$ is indeed $\mathcal{F}_{\tau \wedge T}$-measurable: after $\tau \wedge T$ draws, the agent knows whether she will draw arm 2 for the last $T - \tau \wedge T$ steps or not. With $\mu_1' = \mu_2$, $\mu_2' = \mu_1$, Inequality (4) thus yields:

$$\liminf_{T \to \infty} \frac{\mathbb{E}_\mu\big[\frac{\tau \wedge T}{2}\big]}{\log T} \geq \frac{1}{2\Delta^2/2} .$$

**Unknown gap, ETC strategy:** $\mathcal{S} = \mathcal{H}$. Choosing this time $\mu_1' = (\mu_1 + \mu_2 - \varepsilon)/2$ and $\mu_2' = (\mu_1 + \mu_2 + \varepsilon)/2$, for some $\varepsilon > 0$ yields

$$\liminf_{T \to \infty} \frac{\mathbb{E}_\mu\big[\frac{\tau \wedge T}{2}\big]}{\log T} \geq \frac{1}{\left(\frac{\Delta + \varepsilon}{2}\right)^2} .$$

$R_\mu^\pi(T) = \Delta \mathbb{E}_\mu[N_2(T)]$ for general strategies, while Equation (2) shows that $R_\mu^\pi(T) \geq \Delta \mathbb{E}_\mu[(\tau \wedge T)/2]$ for ETC strategies. Therefore letting $\varepsilon$ go to zero when needed shows that

$$\liminf_{T \to \infty} \frac{R_\mu^\pi(T)}{\log(T)} \geq \frac{C_{\mathcal{S}}^\Pi}{\Delta} , \quad \text{for the value } C_{\mathcal{S}}^\Pi \text{ given at the end of Section 2.}$$

Note that the asymptotic lower bound of Theorem 1 can also be proved by similar arguments, if one really wants to bring in an elephant to kill a mouse. Moreover, note that this proof may also lead to (not-so-simple) non-asymptotic lower-bounds, as shown in Garivier et al. [2016] for example.

## B   Proof of Theorem 1

Let $n \leq T/2$. The number of draws of the suboptimal arm 2 is $N_2 = n + (T - 2n)\mathbb{1}\{S_n \leq 0\}$, where $S_n = (X_1 - Y_1) + \cdots + (X_n - Y_n) \sim \mathcal{N}(n\Delta, 2n)$. The expected regret of the strategy using $2n$ exploration steps is

$$R_\mu^n(T) = \Delta \mathbb{E}_\mu[N_2] = \Delta\big(n + (T - 2n)\mathbb{P}_\mu(S_n \leq 0)\big) . \tag{5}$$

But

$$\mathbb{P}_\mu(S_n \leq 0) = \mathbb{P}_\mu\left(\frac{S_n - n\Delta}{\sqrt{2n}} \leq \frac{-n\Delta}{\sqrt{2n}} = -\Delta\sqrt{\frac{n}{2}}\right) .$$

Denote by $\Phi$ (resp. $\phi$) the pdf (resp. cdf) of the standard Gaussian distribution, and recall that $W$ is the Lambert function defined for all $y > 0$ by $W(y)\exp(W(y)) = y$. The regret is thus upper-bounded as $R_\mu^n(T) \leq \Delta g(n)$ where, for all $x > 0$, $g(x) = x + T\Phi(-\Delta\sqrt{x/2})$. By differentiating $g$, one can see that its maximum is reached at $x^*$ such that

$$\phi\left(\Delta\sqrt{\frac{x^*}{2}}\right) = \frac{\Delta\sqrt{x^*/2}}{T\Delta^2/4} , \quad \text{and thus} \quad x^* = \frac{2}{\Delta^2}W\left(\frac{T^2\Delta^4}{32\pi}\right) .$$

By choosing $\overline{n} = \lceil x^* \rceil$, we obtain

$$R_\mu^{\overline{n}}(T) \leq \frac{2}{\Delta}W\left(\frac{T^2\Delta^4}{32\pi}\right) + \Delta + \Delta T\Phi\left(-\Delta\sqrt{\frac{x^*}{2}}\right) .$$

As $g(\overline{n}) \leq g(x^*) + 1 \leq g(0) + 1 = T/2 + 1$, $R_\mu^{\overline{n}}(T) \leq (T/2 + 1)\Delta \leq 2.04\sqrt{T} + \Delta$ for $T\Delta^2 \leq 4\sqrt{2\pi e}$. If $T\Delta^2 \geq 4\sqrt{2\pi e}$, the inequality $W(y) \leq \log\big((1 + e^{-1})y/\log(y)\big)$ valid for all $y \geq e$ (see Hoorfar and Hassani [2008]) entails

$$W\left(\frac{T^2\Delta^4}{32\pi}\right) \leq \log\left(\frac{(1 + e^{-1})\frac{T^2\Delta^4}{32\pi}}{\log\left(\frac{T^2\Delta^4}{32\pi}\right)}\right) = 2\log\left(\frac{1}{8}\sqrt{\frac{1 + e^{-1}}{\pi}}\frac{T\Delta^2}{\sqrt{\log\left(\frac{T\Delta^2}{4\sqrt{2\pi}}\right)}}\right)$$

In addition, the classic bound on the Gaussian tail $\Phi(-y) \leq \phi(y)/y$ yields by definition of $x^*$:

$$\Phi\left(-\Delta\sqrt{\frac{x^*}{2}}\right) \leq \frac{\phi\left(-\Delta\sqrt{\frac{x^*}{2}}\right)}{\Delta\sqrt{\frac{x^*}{2}}} = \frac{4}{T\Delta^2} .$$

Hence, for $T\Delta^2 \geq 4\sqrt{2\pi e}$,

$$R_\mu^{\overline{n}}(T) \leq \frac{4}{\Delta}\log\left(\frac{e}{8}\sqrt{\frac{1 + e^{-1}}{\pi}}\frac{T\Delta^2}{\sqrt{\log\left(\frac{T\Delta^2}{4\sqrt{2\pi}}\right)}}\right) + \Delta < \frac{4}{\Delta}\log\left(\frac{T\Delta^2}{4.46\sqrt{\log\left(\frac{T\Delta^2}{4\sqrt{2\pi}}\right)}}\right) + \Delta .$$

To complete the proof of the uniform upper-bound, we start from

$$R_\mu^{\overline{n}}(T) \leq \frac{2}{\Delta}W\left(\frac{T^2\Delta^4}{32\pi}\right) + \frac{4}{\Delta} + \Delta .$$

Figure 1: Regret of the Fixed-Budget ETC algorithm with optimal $n$, for $\Delta = 1/5$ (solid line). The linear upper bound $T\Delta/2$ and the logarithmic bound of Theorem 1 are dashed, bold lines. The thin, dotted line is $2.04\sqrt{T} + \Delta$. The bold, dotted line is $\Delta g(\overline{n})$, with $g$ and $\overline{n}$ as in the proof.

Denoting by $r \approx 1.09$ the root of $(4r - 1)W(r) = 2$, the maximum of $2W\left(\frac{T^2\Delta^4}{32\pi}\right)/\Delta + 4/\Delta$ is reached at $\underline{\Delta} = \left(32\pi r/T^2\right)^{1/4}$, and is equal to

$$\frac{2}{\underline{\Delta}}W\left(\frac{T^2\underline{\Delta}^4}{32\pi}\right) + \frac{4}{\underline{\Delta}} = \frac{8r^{3/4}}{(4r-1)(2\pi)^{1/4}}\sqrt{T} < 2\sqrt{T} \ .$$

Let us now prove that choosing $n$ too small leads to catastrophic regret. If $n \leq 4(1-\varepsilon)\log(T)/\Delta^2$, then Equation (5) yields

$$\mathbb{P}_\mu(S_n \leq 0) \geq \frac{1}{\Delta\sqrt{\pi n}}\left(1 - \frac{2}{n\Delta^2}\right)\exp\left(-\frac{\Delta^2 n}{4}\right)$$

$$\geq \left(1 - \frac{2}{n\Delta^2}\right)\frac{1}{2\sqrt{\pi(1-\varepsilon)\log(T)}}\exp\left(-(1-\varepsilon)\log(T)\right)$$

$$\geq \left(1 - \frac{2}{n\Delta^2}\right)\frac{T^{\varepsilon-1}}{2\sqrt{\pi\log(T)}} \ .$$

Thus, the expected regret is lower-bounded as

$$R_\mu^n(T) \geq \Delta(T - 2n)\mathbb{P}_\mu(S_n \leq 0) \geq \Delta\left(1 - \frac{2}{n\Delta^2}\right)\left(1 - \frac{8\log(T)}{\Delta^2 T}\right)\frac{T^\varepsilon}{2\sqrt{\pi\log(T)}} \ .$$

Let us now turn to the last statement of the theorem. The previous inequality shows that for all $\varepsilon > 0$,

$$\liminf_T \inf_{\frac{3}{\Delta^2} < n \leq \frac{4(1-\varepsilon)\log(T)}{\Delta^2}} \frac{R_\mu^n(T)}{\log(T)} = +\infty \ .$$

For $n \le 3/\Delta^2$, we have that

$$\mathbb{P}_\mu(S_n \le 0) \ge \mathbb{P}_\mu\left(\frac{S_n - n\Delta}{\sqrt{2n}} \le -\sqrt{\frac{3}{2}}\right) > 0 \,,$$

and hence that $\liminf_T \inf_{n \le 3/\Delta^2} R_\mu^n(T)/T > 0$. As $R_\mu^n(T) \ge n\Delta$, the result follows.

## C  Proof of Theorem 2

Recall that $\mu_1 > \mu_2$. Using (2), one has

$$R_\mu^\pi(T) \le \Delta\mathbb{E}_\mu\left[\frac{\tau}{2}\right] + T\Delta\,\mathbb{P}_\mu(\tau < T, \hat{a} = 2) \,.$$

If $T\Delta^2 \le 1$, then $\tau = 2$ and $\hat{a}$ is based on a single sample from each action. Therefore $\hat{a} = 2$ with probability less than $1/2$ and the regret is upper-bounded by $\Delta + T\Delta/2$. Otherwise, let $S_0 = 0$, $S_n = (X_1 - Y_1) + \cdots + (X_n - Y_n)$ for every $n \ge 1$. For every $u > 0$, let $n_u = (\log(T\Delta^2) + u)/\Delta^2$. Observe that

$$\left\{\frac{\tau}{2} \ge \left\lceil\frac{\log(T\Delta^2) + u}{\Delta^2}\right\rceil\right\} \subset \left\{S_{\lceil n_u\rceil} \le \frac{\log(T\Delta^2)}{\Delta}\right\} \,.$$

Moreover, if $n_u \in \mathbb{N}$ then $S_{n_u} \sim \mathcal{N}(n_u\Delta, 2n_u)$ and

$$\mathbb{P}_\mu\left(S_{n_u} \le \frac{\log(T\Delta^2)}{\Delta}\right) = \mathbb{P}_\mu\left(\frac{S_{n_u} - n_u\Delta}{\sqrt{2n_u}} \le \frac{\log(T\Delta^2)/\Delta - \Delta(\log(T\Delta^2) + u)/\Delta^2}{\sqrt{2(\log(T\Delta^2) + u)/\Delta^2}}\right)$$

$$= \mathbb{P}_\mu\left(\frac{S_{n_u} - n_u\Delta}{\sqrt{2n_u}} \le \frac{-u}{\sqrt{2(\log(T\Delta^2) + u)}}\right) \le \exp\left(-\frac{u^2}{4\big(\log(T\Delta^2) + u\big)}\right) \,.$$

Hence, for $a = 2\sqrt{\log(T\Delta^2)}$,

$$\int_{n_a}^\infty \mathbb{P}_\mu\left(\frac{\tau}{2} - 1 \ge v\right) dv = \int_a^\infty \mathbb{P}_\mu\left(\frac{\tau}{2} - 1 \ge \frac{\log(T\Delta^2) + u}{\Delta^2}\right)\frac{du}{\Delta^2}$$

$$\le \int_a^\infty \mathbb{P}_\mu\left(\frac{\tau}{2} \ge \left\lceil\frac{\log(T\Delta^2) + u}{\Delta^2}\right\rceil\right)\frac{du}{\Delta^2}$$

$$\le \frac{1}{\Delta^2}\int_a^\infty \exp\left(-\frac{u^2}{4\big(\log(T\Delta^2) + u\big)}\right) du$$

$$\le \frac{1}{\Delta^2}\int_a^\infty \exp\left(-\frac{u}{2\sqrt{\log(T\Delta^2) + 4}}\right) du \qquad \text{as } \log(T\Delta^2) \le u\sqrt{\log(\Delta^2 T)}/2$$

$$\le \frac{1}{\Delta^2}\int_0^\infty \exp\left(-\frac{u}{2\sqrt{\log(T\Delta^2) + 4}}\right) du$$

$$= \frac{2\sqrt{\log(T\Delta^2) + 4}}{\Delta^2} \,,$$

and

$$\mathbb{E}_\mu\left[\frac{\tau}{2}\right] \le 1 + n_a + \int_{n_a}^\infty \mathbb{P}_\mu\left(\frac{\tau}{2} - 1 \ge v\right) dv \le 1 + \frac{\log(T\Delta^2) + 4\sqrt{\log(T\Delta^2)} + 4}{\Delta^2} \,.$$

To conclude the proof of the first statement, it remains to show that $\mathbb{P}(\tau < T, \hat{a} = 2) \le 1/(T\Delta^2)$. Since $X_1 - Y_1 \sim \mathcal{N}(\Delta, 2)$, $\mathbb{E}_\mu[\exp(-\Delta(X_1 - Y_1))] = \exp(-\Delta^2 + 2\Delta^2/2) = 1$ and $M_n = \exp(-\Delta S_n)$ is a martingale. Let $\tau_2 = T \wedge \inf\{n \ge 1 : S_n \le -\log(T\Delta^2)/\Delta\}$, and observe that

$$\left\{\tau < T, \hat{a} = 2\right\} \subset \left\{\exists n < T : S_n \le -\frac{\log(T\Delta^2)}{\Delta}\right\} = \left\{\tau_2 < T\right\} \,.$$

Doob's optional stopping theorem yields $\mathbb{E}_\mu[M_{\tau_2}] = \mathbb{E}_\mu[M_0] = 1$. But as $M_{\tau_2} = \exp(-\Delta S_{\tau_2}) \geq \exp(\Delta \log(T\Delta^2)/\Delta) = T\Delta^2$ on the event $\{\tau_2 < T\}$,

$$\mathbb{P}_\mu(\tau < T, \hat{a} = 2) \leq \mathbb{P}_\mu(\tau_2 < T) \leq \mathbb{E}_\mu\left[\mathbb{1}\{\tau_2 < T\}\frac{M_{\tau_2}}{T\Delta^2}\right] \leq \frac{\mathbb{E}_\mu[M_{\tau_2}]}{T\Delta^2} = \frac{1}{T\Delta^2} \ .$$

The last statement Theorem 2 is obtained by maximising the bound (except for the $\Delta$ summand). Let $u = 1/(\Delta\sqrt{T})$ and $f(u) = -u\log(u/\sqrt{e}) + 2u\sqrt{-\log(u)}$. Denoting $\ell = -\log(u)$, $f'(u) = \ell - 1/2 + 2\sqrt{\ell} - 1/\sqrt{\ell} = (\sqrt{\ell}+2)(\ell - 1/2)/\sqrt{\ell}$, thus the maximum is reached at $\ell = \log\left(\Delta\sqrt{T}\right) = 1/2$ and $\Delta = \sqrt{e/T}$. Re-injecting this value into the bound, we obtain that for every $\Delta > 0$,

$$R_\mu^{\text{SPRT-ETC}}(T) \leq \frac{2 + 4\sqrt{1} + 4}{\sqrt{e/T}} = 10\sqrt{\frac{T}{e}} \ .$$

# D  Proof of Theorem 5

Recall that $\mu_1 = \mu_2 + \Delta$ with $\Delta > 0$. Let $W_s = (X_s - Y_s - \Delta)/\sqrt{2}$, which means that $W_1, W_2, \ldots$ are i.i.d. standard Gaussian random variables. Introducing $\mathcal{F} = (\hat{a} \neq 1, \tau < T)$, one has by (2) that

$$R_\mu^{\text{BAI-ETC}}(T) \leq T\Delta\mathbb{P}_\mu(\mathcal{F}) + (\Delta/2)\mathbb{E}_\mu[\tau \wedge T].$$

From the definition of $\tau$ and Lemma 1.(c) in Appendix G, assuming that $T\Delta^2 \geq 4e^2$, one obtains

$$\begin{aligned}
\mathbb{P}_\mu(\mathcal{F}) &\leq \mathbb{P}_\mu\left(\exists s : 2s \leq T, \ \hat{\mu}_{1,s} - \hat{\mu}_{2,s} \leq -\sqrt{\frac{4}{s}\log\left(\frac{T}{2s}\right)}\right) \\
&= \mathbb{P}_\mu\left(\exists s \leq T/2 : \frac{\sum_{i=1}^s W_i}{s} \leq -\sqrt{\frac{2}{s}\log\left(\frac{T/2}{s}\right)} - \frac{\Delta}{\sqrt{2}}\right) \\
&\leq \frac{120e\sqrt{\log(\Delta^2 T/4)}}{\Delta^2 T} + \frac{64e}{\Delta^2 T} \ .
\end{aligned}$$

The last step is bounding $\mathbb{E}_\mu[\tau \wedge T]$ for which as $T\Delta^2 \geq 4$, Lemma 1.(b) yields

$$\begin{aligned}
\mathbb{E}_\mu[\tau \wedge T] &= \sum_{t=1}^T \mathbb{P}_\mu(\tau \geq t) \leq 2 + 2\sum_{s=1}^{T/2}\mathbb{P}_\mu(\tau \geq 2s + 1) \\
&\leq 2 + 2\sum_{s=1}^{T/2}\mathbb{P}_\mu\left(\frac{\sum_{i=1}^s W_i}{s} \leq \sqrt{\frac{2}{s}\log\left(\frac{T/2}{s}\right)} - \frac{\Delta}{\sqrt{2}}\right) \\
&\leq \frac{8\log\left(\frac{T\Delta^2}{4}\right)}{\Delta^2} + \frac{8\sqrt{\pi\log\left(\frac{T\Delta^2}{4}\right)}}{\Delta^2} + \frac{8}{\Delta^2} + 4 \ ,
\end{aligned}$$

Therefore, if $T\Delta^2 \geq 4e^2$,

$$\begin{aligned}
R_\mu^{\text{BAI-ETC}}(T) &\leq \frac{4\log\left(\frac{T\Delta^2}{4}\right)}{\Delta} + \frac{4\sqrt{\pi\log\left(\frac{T\Delta^2}{4}\right)}}{\Delta} + \frac{4}{\Delta} + 2\Delta + \frac{120e\sqrt{\log\left(\frac{\Delta^2 T}{4}\right)}}{\Delta} + \frac{64e}{\Delta} \\
&\leq \frac{4\log\left(\frac{T\Delta^2}{4}\right)}{\Delta} + \frac{334\sqrt{\log\left(\frac{T\Delta^2}{4}\right)}}{\Delta} + \frac{178}{\Delta} + 2\Delta.
\end{aligned}$$

Taking the limit as $T \to \infty$ shows that $\limsup_{T\to\infty} R_\mu^{\text{BAI-ETC}}(T)/\log(T) \leq 4$. Noting that $R_\mu^{\text{BAI-ETC}}(T) \leq \Delta T$ and taking the minimum of this bound and the finite-time bound given above leads arduously to $R_\mu^{\text{BAI-ETC}}(T) \leq 2\Delta + 32\sqrt{T}$ for all $\mu$.

# E  Proof of Theorem 7

Define random time $\tau = \max\{\tau_1, \tau_2\}$ where $\tau_i$ is given by

$$\tau_i = \min\left\{ t \leq T : \sup_{s \geq t} |\hat{\mu}_{i,s} - \mu_i| < \varepsilon_T \right\}.$$

By the concentration Lemma 1.(a) in Appendix G we have $\mathbb{E}_\mu[\tau_i] \leq 1 + 9/\varepsilon_T^2$ and so $\mathbb{E}_\mu[\tau] \leq \mathbb{E}_\mu[\tau_1 + \tau_2] \leq 2 + 18/\varepsilon_T^2$. For $t > 2\tau$ we have $|\hat{\mu}_{A_{t,\max}}(t-1) - \mu_{A_{t,\max}}| < \varepsilon_T$. Therefore the expected number of draws of the suboptimal arm may be bounded by

$$
\begin{aligned}
\mathbb{E}_\mu[N_2(T)] = \mathbb{E}_\mu\left[ \sum_{t=1}^T \mathbb{1}\{I_t = 2\} \right] &\leq \mathbb{E}_\mu[2\tau] + \mathbb{E}_\mu\left[ \sum_{t=2\tau+1}^T \mathbb{1}\{I_t = 2\} \right] \\
&\leq \mathbb{E}_\mu[2\tau] + \mathbb{E}_\mu\left[ \sum_{t=1}^T \mathbb{1}\left\{ \hat{\mu}_2(t-1) + \sqrt{\frac{2\log(T/N_2(t-1))}{N_2(t-1)}} \geq \mu_1 + \Delta - 3\varepsilon_T \text{ and } I_t = 2 \right\} \right] \\
&\quad + \mathbb{E}_\mu\left[ \sum_{t=1}^T \mathbb{1}\left\{ \hat{\mu}_1(t-1) + \sqrt{\frac{2\log(T/N_1(t-1))}{N_1(t-1)}} \leq \mu_2 + \Delta - \varepsilon_T = \mu_1 - \varepsilon_T \right\} \right] \quad (6)
\end{aligned}
$$

By Lemma 1.(b), whenever $T(2\Delta - 3\varepsilon_T)^2 \geq 2$,

$$
\begin{aligned}
\mathbb{E}_\mu\left[ \sum_{t=1}^T \mathbb{1}\left\{ \hat{\mu}_2(t-1) + \sqrt{\frac{2\log(T/N_2(t-1))}{N_2(t-1)}} \geq \mu_1 + \Delta - 3\varepsilon_T = \mu_2 + 2\Delta - 3\varepsilon_T \text{ and } I_t = 2 \right\} \right] \\
\leq \sum_{s=1}^T \mathbb{P}_\mu\left( \hat{\mu}_{2,s} - \mu_2 + \sqrt{\frac{2\log(T/s)}{s}} \geq 2\Delta - 3\varepsilon_T \right) \\
\leq \frac{2\log\left(\frac{T(2\Delta - 3\varepsilon_T)^2}{2}\right)}{(2\Delta - 3\varepsilon_T)^2} + \frac{2\sqrt{\pi\log\left(\frac{T(2\Delta - 3\varepsilon_T)^2}{2}\right)}}{(2\Delta - 3\varepsilon_T)^2} + \frac{2}{(2\Delta - 3\varepsilon_T)^2} + 1.
\end{aligned}
$$

For the second term in (6) we apply Lemma 1.(c) to obtain, whenever $T\varepsilon_T^2 \geq e^2$,

$$
\begin{aligned}
\mathbb{E}_\mu\left[ \sum_{t=1}^T \mathbb{1}\left\{ \hat{\mu}_1(t-1) + \sqrt{\frac{2\log(T/N_1(t-1))}{N_1(t-1)}} \leq \mu_1 - \varepsilon_T \right\} \right] \\
\leq T\mathbb{P}_\mu\left( \exists s \leq T : \hat{\mu}_{1,s} + \sqrt{\frac{2}{s}\log(T/s)} \leq \mu_1 - \varepsilon_T \right) \leq \frac{30e\sqrt{\log(\varepsilon_T^2 T)}}{\varepsilon_T^2} + \frac{16e}{\varepsilon_T^2}.
\end{aligned}
$$

Therefore, if $T(2\Delta - 3\varepsilon_T)^2 \geq 2$ and $T\varepsilon_T^2 \geq e^2$,

$$
\begin{aligned}
\mathbb{E}_\mu[N_2(T)] \leq{} & \frac{2\log\left(\frac{T(2\Delta - 3\varepsilon_T)^2}{2}\right)}{(2\Delta - 3\varepsilon_T)^2} + \frac{2\sqrt{\pi\log\left(\frac{T(2\Delta - 3\varepsilon_T)^2}{2}\right)}}{(2\Delta - 3\varepsilon_T)^2} + \frac{30e\sqrt{\log(\varepsilon_T^2 T)}}{\varepsilon_T^2} \\
& + \frac{80}{\varepsilon_T^2} + \frac{2}{(2\Delta - 3\varepsilon_T)^2} + 5.
\end{aligned}
$$

The first result follows since the regret is $R_\mu^{\Delta\text{-UCB}}(T) = \Delta\mathbb{E}_\mu[N_2(T)]$. For the second it easily noted that for the choice of $\varepsilon_T$ given in the definition of the algorithm that $\limsup_{T\to\infty} \mathbb{E}_\mu[N_2(T)]/\log(T) \leq 1/(2\Delta^2)$. Therefore $\limsup_{T\to\infty} R_\mu^{\Delta\text{-UCB}}(T)/\log(T) \leq 1/(2\Delta)$. The third result follows from a laborious optimisation step to upper-bound the minimum of $T\Delta$ and the finite-time regret bound above.

# F  Proof of Theorem 8

For any $\varepsilon \in (0, \Delta)$ we have

$$
\begin{aligned}
\mathbb{E}_\mu[N_2(T)] &= \mathbb{E}_\mu\left[\sum_{t=1}^{T} \mathbb{1}\{A_t = 2\}\right] \\
&\leq \mathbb{E}_\mu\left[\sum_{t=1}^{T} \mathbb{1}\left\{A_t = 2 \text{ and } \hat{\mu}_2(t-1) + \sqrt{\frac{2}{N_2(t-1)}\log\left(\frac{T}{N_2(t-1)}\right)} \geq \mu_1 - \varepsilon\right\}\right] \\
&\quad + T\mathbb{P}_\mu\left(\exists s \leq T : \hat{\mu}_{1,s} + \sqrt{\frac{2}{s}\log\left(\frac{T}{s}\right)} \leq \mu_1 - \varepsilon\right).
\end{aligned}
$$

By the concentration Lemma 1.(b) in Appendix G we have, whenever $T(\Delta - \varepsilon)^2 \geq 2$,

$$
\begin{aligned}
\mathbb{E}_\mu&\left[\sum_{t=1}^{T} \mathbb{1}\left\{A_t = 2 \text{ and } \hat{\mu}_2(t-1) + \sqrt{\frac{2}{N_2(t-1)}\log\left(\frac{T}{N_2(t-1)}\right)} \geq \mu_1 - \varepsilon\right\}\right] \\
&\leq \sum_{s=1}^{T} \mathbb{P}_\mu\left(\hat{\mu}_{2,s} - \mu_2 + \sqrt{\frac{2}{s}\log\left(\frac{T}{s}\right)} \geq \Delta - \varepsilon\right) \\
&\leq \frac{2\log\left(\frac{T(\Delta-\varepsilon)^2}{2}\right)}{(\Delta-\varepsilon)^2} + \frac{2\sqrt{\pi\log\left(\frac{T(\Delta-\varepsilon)^2}{2}\right)}}{(\Delta-\varepsilon)^2} + \frac{2}{(\Delta-\varepsilon)^2} + 1
\end{aligned}
$$

For the second term we apply Lemma 1.(c) to obtain, whenever $T\varepsilon^2 \geq e^2$,

$$
T\mathbb{P}_\mu\left(\exists s \leq T : \hat{\mu}_{1,s} + \sqrt{\frac{2}{s}\log\left(\frac{T}{s}\right)} \leq \mu_1 - \varepsilon\right) \leq \frac{30e\sqrt{\log(\varepsilon^2 T)}}{\varepsilon^2} + \frac{16e}{\varepsilon^2}.
$$

Finally, if $T(\Delta - \varepsilon)^2 \geq 2$ and $T\varepsilon^2 \geq e^2$,

$$
R_\mu^{\mathrm{UCB}}(T) \leq \Delta\mathbb{E}_\mu[N_2(T)]
$$

$$
\leq \frac{2\log\left(\frac{T(\Delta-\varepsilon)^2}{2}\right)}{\Delta\left(1 - \varepsilon/\Delta\right)^2} + \frac{2\sqrt{\pi\log\left(\frac{T(\Delta-\varepsilon)^2}{2}\right)}}{\Delta\left(1 - \varepsilon/\Delta\right)^2} + \frac{2}{\Delta\left(1 - \varepsilon/\Delta\right)^2} + \Delta + \Delta\left(\frac{30e\sqrt{\log(\varepsilon^2 T)}}{\varepsilon^2} + \frac{16e}{\varepsilon^2}\right)
$$

The asymptotic result follows by taking the limit as $T$ tends to infinity and choosing $\varepsilon = \log^{-\frac{1}{8}}(T)$ while the minimax result follows by finding the minimum of the finite-time bound given above and the naive $R_\mu^{\mathrm{UCB}}(T) \leq T\Delta$.

# G  Deviation Inequalities

As was already the case in the proof of Theorem 1, we heavily rely on the following well-known inequality on the tail of a Gaussian distribution: if $X \sim \mathcal{N}(0,1)$, then for all $x > 0$

$$
\mathbb{P}(X \geq x) \leq \min\left\{1, \frac{1}{x\sqrt{2\pi}}\right\}\exp\left(-x^2/2\right).
$$

Lemma 1 gathers some more specific results that are useful in our regret analyses, and that we believe to be of a certain interest on their own.

**Lemma 1.** *Let $\varepsilon > 0$ and $\Delta > 0$ and $W_1, W_2, \ldots$ be standard i.i.d. Gaussian random variables and $\hat{\mu}_t = \sum_{s=1}^{t} W_s/t$. Then the following hold:*

*(a).* $\quad \mathbb{E}\left[\min\left\{t : \sup_{s \geq t} |\hat{\mu}_s| \geq \varepsilon\right\}\right] \leq 1 + 9/\varepsilon^2$

*(b).* $\quad$ *if $T\Delta^2 \geq 2$ then* $\displaystyle\sum_{n=1}^{T} \mathbb{P}\left(\hat{\mu}_n + \sqrt{\frac{2}{n}\log\left(\frac{T}{n}\right)} \geq \Delta\right) \leq \frac{2\log\left(\frac{T\Delta^2}{2}\right)}{\Delta^2} + \frac{2\sqrt{\pi\log\left(\frac{T\Delta^2}{2}\right)}}{\Delta^2} + \frac{2}{\Delta^2} + 1$

*(c).* $\quad$ *if $T\varepsilon^2 \geq e^2$ then* $\mathbb{P}\left(\exists s \leq T : \hat{\mu}_s + \sqrt{\frac{2}{s}\log\left(\frac{T}{s}\right)} + \varepsilon \leq 0\right) \leq \frac{30e\sqrt{\log(\varepsilon^2 T)}}{\varepsilon^2 T} + \frac{16e}{\varepsilon^2 T}$

The proof of Lemma 1 follows from standard peeling techniques and inequalities for Gaussian sums of random variables. So far we do not know if this statement holds for subgaussian random variables where slightly weaker results may be shown by adding $\log\log$ terms to the confidence interval, but unfortunately by doing this one sacrifices minimax optimality.

*Proof of Lemma 1.(a).* We use a standard peeling argument and the maximal inequality.

$$
\begin{aligned}
\mathbb{P}\left(\exists s \geq t : |\hat{\mu}_s| \geq \varepsilon\right) &\leq \sum_{k=1}^{\infty} \mathbb{P}\left(\exists s \in [kt, (k+1)t] : |\hat{\mu}_s| \geq \varepsilon\right) \\
&\leq \sum_{k=1}^{\infty} \mathbb{P}\left(\exists s \leq (k+1)t : |s\hat{\mu}_s| \geq kt\varepsilon\right) \\
&\leq \sum_{k=1}^{\infty} 2\exp\left(-\frac{(kt\varepsilon)^2}{2(k+1)t}\right) = \sum_{k=1}^{\infty} 2\exp\left(-\frac{kt\varepsilon^2}{2(1+1/k)}\right) \\
&\leq \sum_{k=1}^{\infty} 2\exp\left(-\frac{kt\varepsilon^2}{4}\right)
\end{aligned}
$$

Therefore

$$
\begin{aligned}
\mathbb{E}[\tau] &\leq 1 + \sum_{t=1}^{\infty} \mathbb{P}\left(\tau \geq t\right) \leq 1 + \sum_{t=1}^{\infty} \min\left\{1, 2\sum_{k=1}^{\infty} \exp\left(-\frac{kt\varepsilon^2}{4}\right)\right\} \\
&= 1 + \sum_{t=1}^{\infty} \min\left\{1, \frac{2}{\exp\left(t\varepsilon^2/4\right) - 1}\right\} \leq 1 + \int_0^{\infty} \min\left\{1, \frac{2}{\exp\left(t\varepsilon^2/4\right) - 1}\right\} dt \\
&\leq 1 + \frac{4\log(4)}{\varepsilon^2} + \int_{4/\varepsilon^2\log(4)}^{\infty} \frac{8}{3\exp\left(t\varepsilon^2/4\right)} dt = 1 + \frac{4\log(4)}{\varepsilon^2} + \frac{8}{3\varepsilon^2} \leq 1 + \frac{9}{\varepsilon^2}.
\end{aligned}
$$

$\square$

*Proof of Lemma 1.(b).* Let $\nu$ be the solution of $\sqrt{2\log(T/n)/n} = \Delta$, that is $\nu = 2W\left(\Delta^2 T/2\right)/\Delta^2$. Then

$$
\sum_{n=1}^{T} \mathbb{P}\left(\hat{\mu}_n + \sqrt{\frac{2}{n}\log\left(\frac{T}{n}\right)} \geq \Delta\right) \leq \nu + \sum_{n=\lceil\nu\rceil}^{T} \mathbb{P}\left(\hat{\mu}_n \geq \Delta - \sqrt{\frac{2}{n}\log\left(\frac{T}{n}\right)}\right).
$$

As for all $n \geq \nu$

$$
\frac{2}{n}\log\frac{T}{n} \leq \frac{2}{\nu}\log\left(\frac{T}{\nu}\right)\frac{\nu}{n} = \Delta^2\frac{\nu}{n},
$$

$$\sum_{n=\lceil\nu\rceil}^{T}\mathbb{P}\left(\hat\mu_n\geq\Delta-\sqrt{\frac{2}{n}\log\left(\frac{T}{n}\right)}\right)\leq\sum_{n=\lceil\nu\rceil}^{\infty}\mathbb{P}\left(\hat\mu_n\geq\Delta\left(1-\sqrt{\frac{\nu}{n}}\right)\right)$$

$$\leq\sum_{n=\lceil\nu\rceil}^{\infty}\exp\left(-\frac{n\Delta^2}{2}\left(1-\sqrt{\frac{\nu}{n}}\right)^2\right)$$

$$=\sum_{n=\lceil\nu\rceil}^{\infty}\exp\left(-\frac{\Delta^2}{2}\left(\sqrt{n}-\sqrt{\nu}\right)^2\right)$$

$$\leq 1+\int_{\nu}^{\infty}\exp\left(-\frac{\Delta^2}{2}\left(\sqrt{x}-\sqrt{\nu}\right)^2\right)dx$$

$$=1+\frac{2}{\Delta}\int_{0}^{\infty}\left(\frac{y}{\Delta}+\sqrt{\nu}\right)\exp\left(-\frac{y^2}{2}\right)dy$$

$$=1+\frac{2}{\Delta^2}+\frac{\sqrt{2\pi\nu}}{\Delta}\ .$$

Hence, as $\nu\leq 2\log(\Delta^2 T/2)/\Delta^2$ whenever $T\Delta^2\geq 2$,

$$\sum_{n=1}^{T}\mathbb{P}\left(\hat\mu_n+\sqrt{\frac{2}{n}\log\left(\frac{T}{n}\right)}\geq\Delta\right)\leq\frac{2}{\Delta^2}\left(\log\left(\frac{T\Delta^2}{2}\right)+1+\sqrt{\pi\log\left(\frac{T\Delta^2}{2}\right)}\right)+1\ .$$

$\square$

*Proof of Lemma 1.(c).* Let $x>0$ and $n\in\mathbb{N}$, then by the reflection principle (eg., Mörters and Peres [2010]) it holds that

$$\mathbb{P}\left(\exists s\leq n:s\hat\mu_s+x\leq 0\right)=2\mathbb{P}\left(n\hat\mu_n+x\leq 0\right)\leq 2\min\left\{\frac{1}{x}\sqrt{\frac{n}{2\pi}},1\right\}\exp\left(-\frac{x^2}{2n}\right)\qquad(7)$$

We prepare to use the peeling technique with a carefully chosen grid. Let

$$\eta=\frac{\log(\varepsilon^2 T)}{\log(\varepsilon^2 T)-1}\quad\text{and}\quad G_k=[\eta^k,\eta^{k+1}[\ .$$

As $T\varepsilon^2>e^2$, one has $\eta\in]1,2[$. Moreover, our choice of $\eta$ leads to the following inequality, that will be useful in the sequel

$$\forall x\geq\varepsilon^{-2},\quad(x/T)^{\frac{1}{\eta}}\leq e\,(x/T)\qquad(8)$$

Using a union bound and then Eq. (7), one can write

$$\mathbb{P}\left(\exists s\leq T:\hat\mu_s+\sqrt{\frac{2}{s}\log\left(\frac{T}{s}\right)}+\varepsilon\leq 0\right)$$

$$\leq\sum_{k=0}^{\infty}\mathbb{P}\left(\exists s\in G_k:s\hat\mu_s+\sqrt{2\eta^k\log\left(1\vee\frac{T}{\eta^{k+1}}\right)}+\eta^k\varepsilon\leq 0\right)$$

$$\leq\sum_{k=0}^{\infty}2\min\left\{1,\sqrt{\frac{\eta}{4\pi}}\frac{1}{\sqrt{\log\left(1\vee\frac{T}{\eta^{k+1}}\right)}+\varepsilon\sqrt{\eta^k/2}}\right\}\left(\frac{\eta^{k+1}}{T}\right)^{\frac{1}{\eta}}\exp\left(-\frac{\eta^{k-1}\varepsilon^2}{2}\right)$$

$$\leq\sum_{k=0}^{\infty}f(k)\leq 2\max_{k}f(k)+\int_{0}^{\infty}f(u)du\ ,$$

where the function $f$ is defined on $[0,+\infty[$ by

$$f(u)=2\min\left\{1,\sqrt{\frac{\eta}{4\pi\log\left(\frac{T}{\eta^{u+1}}\right)}}\right\}\left(\frac{\eta^{u+1}}{T}\right)^{\frac{1}{\eta}}\exp\left(-\frac{\eta^{u-1}\varepsilon^2}{2}\right),$$

with the convention that $\sqrt{\frac{\eta}{4\pi \log\left(\frac{T}{\eta^{u+1}}\right)}} = +\infty$ for $u$ such that $T/\eta^{u+1} < 1$.

The last inequality relies on the fact that $f$ can be checked to be unimodal, which permits to upper bound the sum with an integral. The maximum of $f$ is easily upper bounded as follows, using notably (8):

$$\max_k f(k) \le 2 \sup_{k \ge 0} \left(\frac{\eta^{k+1}}{T}\right)^{\frac{1}{\eta}} \exp\left(-\frac{\eta^{k-1}\varepsilon^2}{2}\right) = 2\exp(-1/\eta)\left(\frac{2\eta}{\varepsilon^2 T}\right)^{\frac{1}{\eta}} \le \frac{8e}{\varepsilon^2 T}.$$

The remainder of the proof is spent bounding the integral, which will be split into three disjoint intervals with boundaries at constants $k_1 < k_2$ given by

$$k_1 = \log(1/\varepsilon^2)/\log(\eta) \qquad\qquad k_2 = 1 + \frac{\log\left(\frac{-\log\log(\eta)}{\varepsilon^2}\right)}{\log(\eta)}.$$

These are chosen such that

$$\eta^{k_1} = \varepsilon^{-2} \qquad\qquad \exp\left(-\frac{\eta^{k_2-1}\varepsilon^2}{2}\right) = \sqrt{\log(\eta)}.$$

First, one has

$$I_1 := \int_0^{k_1} f(u)du \le \int_0^{k_1} \frac{\sqrt{2}}{\sqrt{\pi \log\left(\frac{T}{\eta^{u+1}}\right)}} \left(\frac{\eta^{u+1}}{T}\right)^{\frac{1}{\eta}} \exp\left(-\frac{\eta^{u-1}\varepsilon^2}{2}\right) du$$

$$\le \frac{\sqrt{2}}{\sqrt{\pi \log\left(\frac{T}{\eta^{k_1+1}}\right)}} \int_0^{k_1} \left(\frac{\eta^{u+1}}{T}\right)^{\frac{1}{\eta}} du \le \frac{\sqrt{2}}{\sqrt{\pi \log\left(\frac{\varepsilon^2 T}{\eta}\right)}} \frac{\eta}{\log(\eta)} \left(\frac{\eta}{\varepsilon^2 T}\right)^{\frac{1}{\eta}}$$

$$\le \frac{e\sqrt{2}\eta^2}{\sqrt{\pi \log\left(\frac{\varepsilon^2 T}{\eta}\right)} \log(\eta)\varepsilon^2 T} \le \frac{e\sqrt{2}\eta^2}{\sqrt{\pi \log\left(\frac{\varepsilon^2 T}{\eta}\right)}} \frac{\log(\varepsilon^2 T)}{\varepsilon^2 T},$$

where the last inequality follows from the fact that $(\log(\eta))^{-1} \le \eta/(\eta-1) = \log\left(\varepsilon^2 T\right)$.

Secondly,

$$I_2 := \int_{k_1}^{k_2} f(u)du \le \int_{k_1}^{k_2} \frac{\sqrt{2}}{\sqrt{\pi \log\left(\frac{T}{\eta^{u+1}}\right)}} \left(\frac{\eta^{u+1}}{T}\right)^{\frac{1}{\eta}} \exp\left(-\frac{\eta^{u-1}\varepsilon^2}{2}\right) du$$

$$\le \frac{e\sqrt{2}}{\sqrt{\pi \log\left(\frac{T}{\eta^{k_2+1}}\right)}} \int_{k_1}^\infty \frac{\eta^{u+1}}{T} \exp\left(-\frac{\eta^{u-1}\varepsilon^2}{2}\right) du = \frac{e\sqrt{2}}{\sqrt{\pi \log\left(\frac{T}{\eta^{k_2+1}}\right)}} \frac{2\eta^2 \exp\left(-\frac{\eta^{k_1-1}\varepsilon^2}{2}\right)}{\varepsilon^2 T \log(\eta)}$$

$$\le \frac{2e\sqrt{2}}{\sqrt{\pi \log\left(\frac{T}{\eta^{k_2+1}}\right)}} \frac{\eta^2}{\varepsilon^2 T \log(\eta)} \le \frac{2e\sqrt{2}\eta^2}{\sqrt{\pi \log\left(\frac{\varepsilon^2 T}{-\eta^2 \log\log(\eta)}\right)}} \frac{\log(\varepsilon^2 T)}{\varepsilon^2 T}$$

$$\le \frac{2e\sqrt{2}\eta^2}{\sqrt{\pi \log\left(\frac{\varepsilon^2 T}{\eta^2 \log\log(\varepsilon^2 T)}\right)}} \frac{\log(\varepsilon^2 T)}{\varepsilon^2 T}.$$

The second inequality follows from (8), since by definition of $k_1$, one has $\eta^{u+1} \ge \varepsilon^{-2}$ for $u \ge k_1$, whereas the last inequalities use again that $(\log(\eta))^{-1} \le \log(\varepsilon^2 T)$. Using similar arguments for the

third term, one has

$$I_3 := \int_{k_2}^{\infty} f(u)du \leq 2 \int_{k_2}^{\infty} \left(\frac{\eta^{u+1}}{T}\right)^{\frac{1}{\eta}} \exp\left(-\frac{\eta^{u-1}\varepsilon^2}{2}\right) du$$

$$\leq 2e \int_{k_2}^{\infty} \frac{\eta^{u+1}}{T} \exp\left(-\frac{\eta^{u-1}\varepsilon^2}{2}\right) du = \frac{4e\eta^2 \exp\left(-\frac{\eta^{k_2-1}\varepsilon^2}{2}\right)}{\varepsilon^2 T \log(\eta)}$$

$$= \frac{4e\eta^2}{\varepsilon^2 T \sqrt{\log(\eta)}} \leq \frac{4e\eta^2}{\varepsilon^2 T} \sqrt{\log(\varepsilon^2 T)}$$

Combining the three upper bounds yield

$$\int_0^{\infty} f(u)du \leq I_1 + I_2 + I_3$$

$$\leq \frac{e\eta^2 \log(\varepsilon^2 T)}{\varepsilon^2 T} \left( \frac{\sqrt{2}}{\sqrt{\pi \log\left(\frac{\varepsilon^2 T}{\eta}\right)}} + \frac{2\sqrt{2}}{\sqrt{\pi \log\left(\frac{\varepsilon^2 T}{\eta^2 \log\log(\varepsilon^2 T)}\right)}} + \frac{4}{\sqrt{\log(\varepsilon^2 T)}} \right)$$

$$\leq \frac{4e \log(\varepsilon^2 T)}{\varepsilon^2 T} \left( \frac{\sqrt{2}}{\sqrt{\pi \log\left(\frac{\varepsilon^2 T}{2}\right)}} + \frac{2\sqrt{2}}{\sqrt{\pi \log\left(\frac{\varepsilon^2 T}{4 \log\log(\varepsilon^2 T)}\right)}} + \frac{4}{\sqrt{\log(\varepsilon^2 T)}} \right)$$

It can be shown, using notably the inequality $\log u \leq u/e$, that for all $\varepsilon^2 T \geq e^2$,

$$\log(\varepsilon^2 T/2) \geq (1/2) \log(\varepsilon^2 T)$$

$$\log\left(\frac{\varepsilon^2 T}{4 \log\log(\varepsilon^2 T)}\right) \geq \left(1 - \frac{4}{e^2}\right) \log(\varepsilon^2 T)$$

and one obtains

$$\int_0^{\infty} f(u)du \leq \frac{4e \log(\varepsilon^2 T)}{\varepsilon^2 T} \times \frac{2 + 2e\sqrt{2}/(\sqrt{e^2 - 4}) + 4\sqrt{\pi}}{\sqrt{\pi \log(\varepsilon^2 T)}} \leq \frac{30e}{\varepsilon^2 T} \sqrt{\log(\varepsilon^2 T)}.$$

Finally,

$$\mathbb{P}\left(\exists s \leq T : \hat{\mu}_s + \sqrt{\frac{2}{s} \log\left(\frac{T}{s}\right)} + \varepsilon \leq 0\right) \leq \int_0^{\infty} f(u)du + 2 \max_k f(k) \leq \frac{30e}{\varepsilon^2 T} \sqrt{\log(\varepsilon^2 T)} + \frac{16e}{\varepsilon^2 T}.$$

$\square$