[Reviews · NeurIPS 2016]

Reviewer 1

Summary

This paper studies a two arm bandit problem with Gaussian reward distributions. This simple setting is used to illustrate the sub-optimality of policies that use two separate phases: an exploration phase during which the algorithm experiments, and an exploitation phase during which it sticks to the with best performance during the exploration phase. They focus on two settings -- with known and unknown gap between the Gaussian distributions -- and three approaches -- an explore-then-commit policy with non-adaptive stopping rule, and ETC policy with adaptive stopping rule, and a fully adaptive allocation rule. The main lesson is that more adaptive rules have better performance.

Qualitative Assessment

This paper highlights the benefit of adaptive algorithms in a transparent manner, and I think it will be valuable for the NIPS community. The advantage of adaptivity in this paper shows up only in constant multiplicative factors, but these are important in practice, and would be missed by anything but a very careful analysis. I gave the paper scores of 3, but I don't view it as borderline. I would accept this paper. Given that the goal illustrating the benefits of adaptivity, I'd suggest providing more intuition for the results. Really, it's quite natural that if you commit to playing arm 1 for times t > tau, but subsequent observations suggest arm 1 is bad, you'd wish you could switch back to arm 2. The track and stop rule in section 6 comes out of an optimization problem that aims to minimize the number of samples required to accumulate a given amount of information. It's pretty intuitive for there to be a gap between this, and the optimization problem solved by an optimal bandit algorithm, which essentially minimizes the regret incurred in accumulating a given amount of information. You don't need to include comments of this form, but they might extend the impact of the paper. As the paper acknowledges, a great deal is known about each of the settings considered due to classic work in sequential statistics by Wald, Robbins, Katehakis, Lai, and likely others. Most importantly, it's known from Wald that adaptive stopping rules offer equivalent statistical power with a smaller expected sample size than any non-adaptive rule. This work is cited on page 4, but anyone who simply read the summary of results might not realize that this important insight should really be attributed to Wald. I think best practice would be to highlight this in the intro.

Confidence in this Review

3-Expert (read the paper in detail, know the area, quite certain of my opinion)


Reviewer 2

Summary

The paper analyzes a two armed bandit problem with normally distributed rewards. The objective of the paper is to show the sub optimality of fixed design or explore and then commit (ETC) strategies compared to fully adaptive strategies. To that end, they establish lower and upper bounds in the three settings to show the improvements from one to the other.

Qualitative Assessment

I found the paper interesting. While I do not find surprising that fully adaptive strategies necessarily improve on fixed design or ETC strategies, I find it very elegant that the authors are able to crisply quantify the improvement through the constant that precedes the logarithmic factor in the regret. While the appeal may not be as broad for an oral level presentation, I certainly support the submission for a poster presentation.

Confidence in this Review

2-Confident (read it all; understood it all reasonably well)


Reviewer 3

Summary

The paper is concerned with obtaining very precise upper and lower bounds on the regret of algorithms for the two-armed bandit problem with normally distributed rewards. The paper considers two settings: one in which the gap between the mean payoffs of the two arms is known, and another in which the gap is unknown. For both settings, the main result compares the asymptotically optimal regret bound for "explore-then-commit" (ETC) strategies --- those which engage in uniform exploration until a stopping time is reached and then commit to the empirically superior arm at that time --- against the asymptotically optimal bound for fully sequential policies. The latter class of policies is found to improve the regret of ETC policies by a factor of 2, asymptotically, in both the fixed-gap and unknown-gap settings. In the fixed-gap setting, this involves proving new upper and lower bounds both for ETC strategies and for fully sequential policies. In the unknown-gap setting, it involves proving new upper and lower bounds for ETC strategies, and a mildly novel result about fully sequential strategies whose novelty lies in simultaneously attaining asymptotically optimal regret and order-optimal minimax regret.

Qualitative Assessment

I really like this paper! It delivers a crisp, clear message --- fully sequential policies are better than ETC by a factor of 2 --- which is important because of the widespread use of ETC policies in A/B testing. I didn't read all of the proofs in detail, but the technical work required to obtain this result appears pretty substantial.

Confidence in this Review

2-Confident (read it all; understood it all reasonably well)


Reviewer 4

Summary

This theoretical paper investigates the three main classes of algorithm to address the 2-armed bandit problem when the time horizon T is known: 1 the fixed-design algorithms, where the stopping time of the exploration phase is fixed in advance, 2 the explore then commit algorithms, where the stopping time of the exploration phase is a random variable, 3 the fully sequential algorithms, where exploration and exploitation are done simultaneously. This paper provides new analytical results on lower bounds of the three classes of algorithms. In particular when the gap is unknown, the authors show that explore then commit algorithms suffer from an asymptotic regret two times higher than the one of fully sequential algorithms. Algorithms and fine analysis of regret upper bounds are given for each class when the gap is known and unknown.

Qualitative Assessment

The main analytical result is significant: fully sequential algorithms are better for regret minimization than explore then commit algorithms. This result was empirically known, while the lower bounds were roughly the same. This paper provides the proof in the case of 2-armed bandits, that explore then commit algorithms suffer from an asymptotic regret two times higher than the one of fully sequential algorithms. It should interest the community. My concern is the presentation. On the form, the abstract claims that "empirical evidence that theory also holds in practice", while empirical results are not given in the paper, but in the appendix. The paper has to respect the format. On the substance, the discussion suggests that the results hold also for K-armed bandits, and then that explore-then-commit algorithms are sub-optimal for the regret minimization problem. This should be true. Usually explore-then-commit and fixed-design algorithms are used for the best arm identification problem. Median Elimination (a fixed-design algorithm) is optimal, while Successive Elimination (an explore-then-commit algorithm) is not optimal, while fully sequential algorithms do not output the best arm. In practice Median Elimination does not work well for regret minimization while Successive Elimination works well. So explore-then-commit algorithm such as Successive Elimination is a reasonable suboptimal choice for regret minimization and best arm identification problems, while fixed-design and fully sequential algorithms are optimal on one problem and inefficient on the other. This seems somewhat contradictory with your claims ? Ok. I moved my notation. Thank you to take into account my concern on presentation.

Confidence in this Review

2-Confident (read it all; understood it all reasonably well)


Reviewer 5

Summary

This paper aims at providing theoretical evidence to support the fact that Explore-Then-Commit strategies are necessarily sub-optimal in terms of regret in a multi-armed bandit setting. In this regard the paper considers a simple two armed bandit problem with Gaussian rewards with different means but same variance. In this setting, the authors show lower bounds for ETC strategies. This shows that ETC strategies are necessarily sub-optimal compared to some strategies that follow fully sequential exploration. The paper also shows that a strategy similar to UCB* is asymptotically optimal and order optimal in the minimax sense at the same time.

Qualitative Assessment

1. One interesting part of the paper to me is the strategy in Algorithm 5 that is asymptotically optimal and optimal in the minimax sense. Although the strategy is not very original (a small modification of a strategy suggested by Lai), it is a new observation for a strategy to be asymptotically optimal and minimax optimal. However, it should be noted that it is optimal only in the case of two arms, which is a major drawback. The authors are encouraged to improve on this result. 2. I like the idea of using the hypothesis test result of Wald to come up with the ETC strategy in Algorithm 2 that achieves the lower bound for ETC strategy. This idea is new to the best of my knowledge. However, again it is challenging to extend this to multi-armed case. 3. I feel that Theorem 5 is rather straight forward to see from the results of the optimal BAI algorithm and there is not a lot of novelty in this result. 4. Although presented as the main result of this paper, I do not see very novel ideas in the proof of the lower bounds for these strategies. Hypothesis testing inequalities stemming from a particular inequality in Tsybakov's book " Introduction to Nonparametric Estimation" , has been used before in the bandit literature for proving lower bounds ". In fact the main step in the proof is Garivier's inequality. The parts of the proof that follows is fairly straight forward. Moreover the results have been presented only for the two arm setting. 5. Overall I like the flavor of the results presented in the paper. I like the Algorithm 5 and corresponding proof, as to the best of my knowledge this is original in the literature. The lower bounds for ETC are new results but may seem straight forward. The major drawback of the paper is the simple two arm setting and that there is not much practical significance of the paper. I would encourage the authors to generalize these results in future. 6. The presentation of the paper is very clear.

Confidence in this Review

2-Confident (read it all; understood it all reasonably well)


Reviewer 6

Summary

This paper study the explore-then-commit strategies in the multi-armed bandit setting. The purpose of this article is to show that the explore-then-commit strategy based exportation and exploration is necessarily suboptimal. As the authors mentioned recent progress on optimal exploration strategies have suggested that well-tuned variants of two-phase strategies might be near-optimal. This paper On the contrary, show that optimal strategies for multi-armed bandit problems must be fully-sequential, and in particular should mix exploration and exploitation. This paper proved the lower bound for the Explore-Then-Commit strategy. And showed that a fully sequential strategy can be better, which proved that ECT are necessarily sub-optimal. The main contribution of this paper is to fully characterize the achievable asymptotic regret when the gap is either known or unknown and the strategies are either fixed-design, ETC or fully sequential.

Qualitative Assessment

The argument made about the explore-then-commit strategy in multi-armed bandit is very interesting and novel. And the authors also proved that this strategy is necessarily suboptimal by providing lower bounds for ETC strategy and general strategy with both known and unknown gap. Also empirical results also provide evidence for the conclusion to hold in practice. One important limitation of this work is that the main proof in this paper is focusing on the two-armed bandit case. The extension to multi-armed bandit cases is not straightforward and is not clearly addressed.

Confidence in this Review

2-Confident (read it all; understood it all reasonably well)